# Identification of pathways to high-level vancomycin resistance in *Clostridioides difficile* that incur high fitness costs in key pathogenicity traits

**Jessica E. Buddle[1], Lucy M. Thompson[1], Anne S. Williams[2], Rosanna C. T. Wright[3], William M. Durham[2], Claire E. Turner[1], Roy R. Chaudhuri[1], Michael A. Brockhurst[3]\*, Robert P. Fagan** [1]\*

1 Molecular Microbiology, School of Biosciences, University of Sheffield, Sheffield, United Kingdom, 2 Department of Physics and Astronomy, University of Sheffield, Sheffield, United Kingdom, 3 Division of Evolution and Genomic Sciences, University of Manchester, Manchester, United Kingdom

\* michael.brockhurst@manchester.ac.uk (MAB); r.fagan@sheffield.ac.uk (RPF)

## Abstract

*Clostridioides difficile* is an important human pathogen, for which there are very limited treatment options, primarily the glycopeptide antibiotic vancomycin. In recent years, vancomycin resistance has emerged as a serious problem in several gram-positive pathogens, but high-level resistance has yet to be reported for *C. difficile*, although it is not known if this is due to constraints upon resistance evolution in this species. Here, we show that resistance to vancomycin can evolve rapidly under ramping selection but is accompanied by fitness costs and pleiotropic trade-offs, including sporulation defects that would be expected to severely impact transmission. We identified 2 distinct pathways to resistance, both of which are predicted to result in changes to the muropeptide terminal D-Ala-D-Ala that is the primary target of vancomycin. One of these pathways involves a previously uncharacterised D,D-carboxypeptidase, expression of which is controlled by a dedicated two-component signal transduction system. Our findings suggest that while *C. difficile* is capable of evolving high-level vancomycin resistance, this outcome may be limited clinically due to pleiotropic effects on key pathogenicity traits. Moreover, our data identify potential mutational routes to resistance that should be considered in genomic surveillance.

## Introduction

*Clostridioides difficile* is the most common cause of antibiotic-associated diarrhoea worldwide, resulting in significant morbidity and mortality [1] that places a huge burden on healthcare systems [2,3]. Most cases of nosocomial *C. difficile* infection (CDI) follow recent antibiotic treatment, which, through altering microbial diversity in the colon, reduces microbiota-mediated colonisation resistance [4]. Although restoring microbial community diversity through faecal microbiota transplantation (FMT) is a potential future treatment for CDI [5], current

**Data Availability Statement:** Genome sequence data for parental strains, P30 resistant isolates and

respective controls, as well as pooled population sequencing data for resistant populations and controls at P10, 20 and 30 is deposited with the European Nucleotide Archive (ENA) under study accession number PRJEB66266. All other data are contained within the paper and Supporting Information files.

**Funding:** JEB is supported by a studentship from the UK Medical Research Council Discovery Medicine North (DiMeN) Doctoral Training Partnership (MR/R015902/1), with Summit Therapeutics Inc as an industrial partner. RCTW is supported by the UK Biotechnology and Biological Sciences Research Council grant (BB/T014342/1) and MAB by a Wellcome Trust Collaborative Award (220243/Z/20/Z). CET is a Royal Society & Wellcome Trust Sir Henry Dale Fellow (208765/Z/17/Z). The funders had no role in study design, data collection and analysis, decision to publish, or preparation of the manuscript.

**Competing interests:** Summit Therapeutics Inc were industrial partners for JEB's MRC DiMeN iCASE PhD studentship but had no input in study design, interpretation or manuscript preparation. The authors declare no further competing interests.

**Abbreviations:** AUC, area under curve; BHI, brain heart infusion; CDI, *C. difficile* infection; CFU, colony-forming unit; FMT, faecal microbiota transplantation; LB, Luria–Bertani; MIC, minimum inhibitory concentration; PCA, principal component analysis; TY, tryptone yeast.

treatment relies on additional antibiotics, most commonly metronidazole or vancomycin. While these can resolve the CDI, they further exacerbate damage to the microbiota, leading to recurrent CDI in up to 25% of cases [6]. Due to increasing incidence of resistance and consequently poor patient outcomes, use of metronidazole has declined in recent years as a result of changing clinical guidelines [7,8]; vancomycin is now the recommended front line antibiotic in the United Kingdom [9] and has been widely adopted worldwide [10].

Vancomycin is a glycopeptide antibiotic that binds to the terminal D-Ala-D-Ala residues on peptidoglycan muropeptide precursors, sterically blocking transglycosylation and transpeptidation reactions [11]. Although relatively slow to emerge, resistance to vancomycin is now found globally in *Staphylococcus aureus* (VRSA) and is widespread in several *Enterococcus* spp. (VRE) [12,13]. Vancomycin resistance in Enterococci is usually associated with one of several *van* gene clusters which encode the enzymes that modify peptidoglycan to remove the vancomycin binding site. Here, the D-Ala-D-Ala is replaced with either D-Ala-D-Lac (e.g., *vanA*, also seen in VRSA), conferring high-level resistance, or D-Ala-D-Ser (e.g., *vanG*), conferring low-level resistance [14].

Vancomycin susceptibility of *C. difficile* is not routinely tested in clinical laboratories making monitoring the emergence of resistance extremely challenging. However, the effectiveness of vancomycin against CDI has declined over time [15] and multiple case reports show reduced vancomycin susceptibility in individual clinical isolates [16]. Although a complete *vanG* cluster is found in diverse *C. difficile* strains [17], whether this confers vancomycin resistance is unclear [18]. However, increased expression of the *vanG* cluster does appear to be associated with reduced susceptibility to vancomycin; mutations in the *vanSR*-encoded two-component system, reducing vancomycin susceptibility through derepression of the *vanG* cluster, have been reported in both clinical isolates and laboratory evolution experiments [19,20]. Beyond such regulatory changes of the *vanG* cluster, the molecular mechanisms underpinning the evolution of vancomycin resistance in *C. difficile* remain unknown. For example, we do not know if other mutations occurring within the *vanG* cluster or at other loci in the *C. difficile* genome contribute to increasing resistance observed clinically. Moreover, whether the evolution of vancomycin resistance is associated with pleiotropic phenotypic effects or fitness costs that might limit the emergence and spread of resistance is poorly understood in *C. difficile*.

To understand the evolutionary dynamics and molecular mechanisms of vancomycin resistance in *C. difficile*, we experimentally evolved 10 replicate populations at increasing concentrations of vancomycin. Within just 250 generations, we observed the evolution of 16- to 32-fold increased vancomycin minimum inhibitory concentration (MIC). To identify the causal genetic variants, we genome sequenced both endpoint-resistant clones and whole populations at multiple time points during the evolution experiment, and then reintroduced key mutations observed in the evolved resistant lineages into the wild-type ancestral genetic background. Evolution of increased vancomycin resistance was associated with mutations in 2 distinct pathways, occurring either in *vanT* within the *vanG* cluster or in a gene encoding a regulator of a previously uncharacterised D,D-carboxypeptidase. Mutations in either pathway are likely to lead to modification of the vancomycin target in the cell wall peptidoglycan and each was associated with defects in growth and sporulation of varying severity. Together, our results propose a new mechanistic model for vancomycin resistance emergence in *C. difficile*, potentially expanding the genetic determinants of resistance that should be monitored in clinical genomic epidemiology. Moreover, our data suggest that the initial emergence of vancomycin resistance in *C. difficile* in the clinic may be severely constrained by fitness trade-offs and pleiotropic effects upon key transmission and virulence traits.

## Results

### Vancomycin resistance evolves rapidly in *C. difficile* during in vitro experimental evolution

We first generated genetically barcoded ancestral strains in an avirulent background with either a wild-type or an elevated mutation rate. Specifically, R20291, a clinically relevant ribotype 027 *C. difficile* strain, was rendered avirulent through complete deletion of 18 kb spanning the entire PaLoc that includes the genes encoding both major toxins and associated regulatory proteins, creating strain R20291Δ*PaLoc*. A subsequent deletion, removing the *mutSL* genes encoding a DNA-damage repair system, generated a hyper-mutable variant R20291Δ*PaLoc*Δ-*mutSL*, with an approximately 20-fold higher mutation rate than the wild type. Five distinct derivatives of each ancestral strain were then generated through introduction of a 9-nucleotide barcode sequence downstream of the *pyrE* gene, resulting in 10 individually barcoded replicate lines used in the evolution experiment (R20291Δ*PaLoc pyrE*::barcode 1–5; R20291Δ*PaLoc*Δ-*mutSL pyrE*::barcode 7–11). Each of the 10 barcoded strains was used to inoculate a 6-well plate containing media supplemented with vancomycin at 0.25×, 0.5×, 1×, 2×, 4×, and 8× the initial MIC of 1 µg/ml. Populations were passaged every 48 h, whereby cells from the well with the highest antibiotic concentration permitting growth were propagated in a 1:400 dilution to a fresh 6-well plate. This process was repeated for a total of 30 serial transfers per replicate line, with adjustment of the vancomycin gradient over time as the growth-permitting vancomycin concentration in each evolving line increased. Ten corresponding control populations were propagated under equivalent conditions in the absence of vancomycin. Populations underwent approximately 8.64 generations per transfer, yielding approximately 259 generations throughout the course of the experiment.

Resistance evolved rapidly in all 10 replicate populations propagated with vancomycin selection (Fig 1A). Nine of the 10 replicate lines evolved to grow in 2 µg/ml vancomycin (an apparent MIC of 4 µg/ml, the EUCAST breakpoint) by the end of the second passage (P2) and all 10 grew in the presence of 8 µg/ml (Bc2, 3, 4) or 16 (Bc1, 5, 7, 8, 9, 10, 11) µg/ml vancomycin by P30 (Fig 1A). The emergence of high-level resistance was significantly accelerated in the hyper-mutable replicate lines (Fig 1B). At least 6 individual clones were isolated from each evolved population at the endpoint and their vancomycin MIC was determined. Of the 82 clones tested, 38 (46%) had an MIC consistent with the vancomycin concentration permitting growth of the population from which they were isolated, while the remainder had an MIC slightly lower than expected [for 42 clones their MIC was 2-fold lower, whereas for 2 clones their MIC was 4-fold lower than the vancomycin concentration permitting growth of the population from which they were isolated (S1 Data)], demonstrating that significant variation still existed within some evolved populations by P30.

### Genetic bases of evolved vancomycin resistance

To understand the genetic bases of increased vancomycin resistance in individual endpoint clones, one clone per replicate population that had an MIC representative of their population was selected for whole genome sequencing. The parental strains for each barcoded lineage, and one random endpoint clone from each replicate control population, were also sequenced. For analyses, we focused on mutations that were observed in the vancomycin-treated lines but never in ancestral or in control evolved clones because these are the most likely to have evolved in response to vancomycin selection. Sequence variants unique to the vancomycin-evolved clones were identified using Varscan [21], validated with Breseq and IGV [22] after mapping to the reference R20291 genome (Fig 1C and S2 Data). We observed between 1 and 3 unique

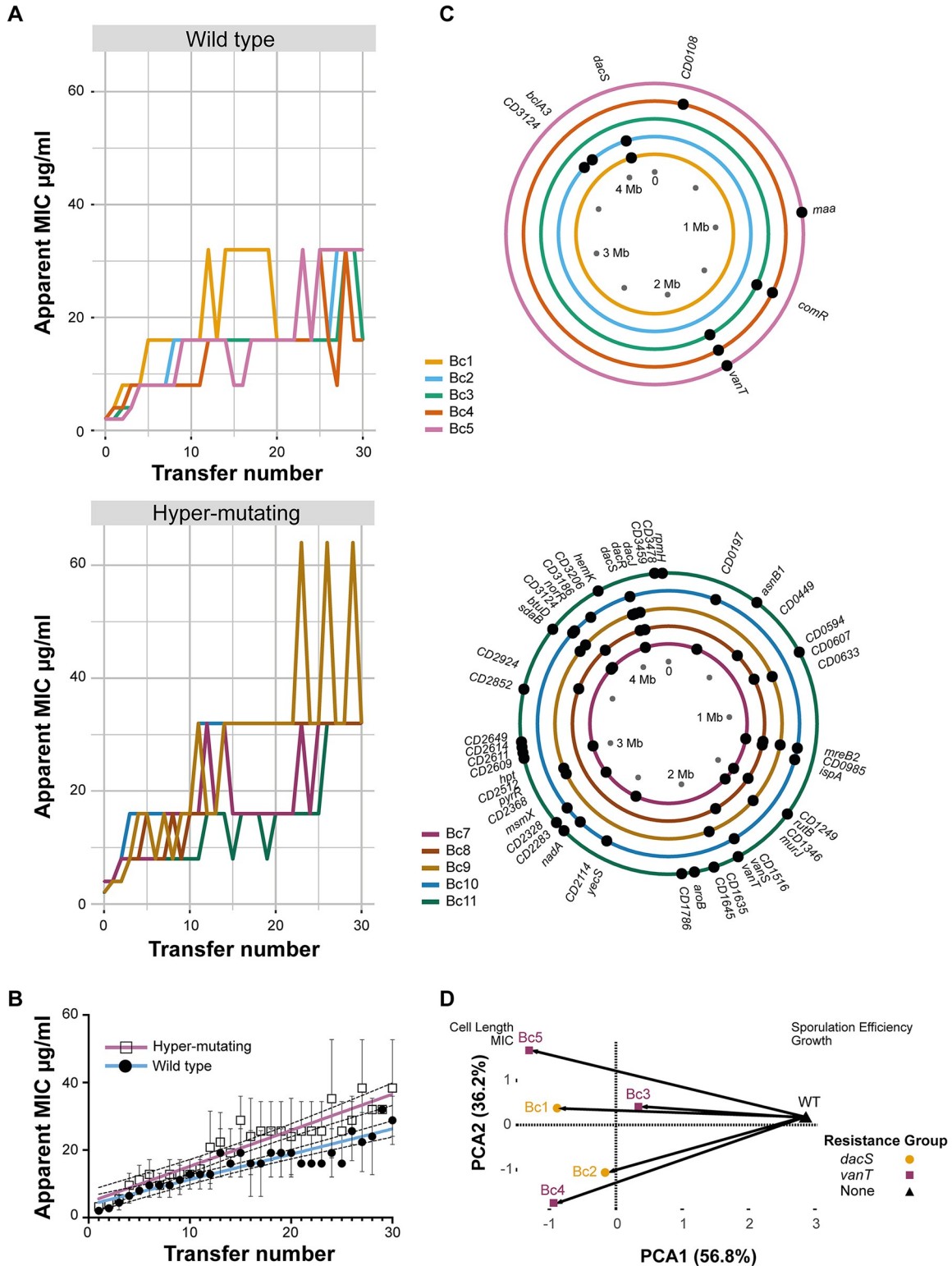

**Fig 1. Evolution of vancomycin resistance.** (**A**) Changes in apparent vancomycin MIC over the course of a 30 transfer experimental evolution. MIC was determined as the well with the lowest vancomycin concentration showing no clear growth. (**B**) Shown are the means and standard deviations of the apparent MIC for 5 wild-type (open squares) and 5 hyper-mutating (closed circles) populations. Linear regressions fitted to each data set, blue and pink, respectively, are significantly different by ANCOVA, $P = 0.0008$. The data underlying panels **A** and **B** can be found in S5 Data. (**C**) Shown are the chromosomal locations of non-synonymous variant alleles in

isolated wild-type (top) and hyper-mutating (bottom) *C. difficile* endpoint clones, excluding any mutations that were also identified in any of the control strains. Each circle represents a single *C. difficile* genome, colour coded according to population as indicated in the key on the left. The data underlying this panel, including a full list of all variants shown here along with synonymous and intergenic mutations, can be found in S2 Data. (**D**) PCA of P30 isolates from populations Bc1-5 (coloured points) vs. the ancestral strain (black triangle), with PC1 vs. PC2 plotted, accounting for 93% variance. The loadings (sporulation efficiency, growth, MIC, cell length) are shown in respective locations. Arrows show the evolutionary trajectories of wild-type replicate lines from their ancestor in multivariate phenotype space. The data underlying this figure can be found in S5 Data. MIC, minimum inhibitory concentration; PCA, principal component analysis.

mutations per genome in wild-type evolved clones and between 14 and 26 in hyper-mutator evolved clones from the vancomycin treatment. Genic SNPs accounted for 43% of the variants, of which 67.4% were nonsynonymous and 32.6% were synonymous, with frameshifts accounting for a further 31% of these variants.

Parallel evolution, where mutations affecting the same locus arise in multiple independently evolving replicate populations, is strong evidence for the action of selection and suggests a potential role for these mutations in adaptation. In the 5 wild-type vancomycin selected lines, we observed parallel evolution occurring at 3 genomic loci: *vanT* in 3 clones, *CDR20291_3437* (simplified to *CD3437* hereafter) in 2 clones, and *comR* in 2 clones. Whereas mutations in *vanT* and *CD3437* were mutually exclusive in wild-type evolved clones, mutations in *comR* always co-occurred with mutations in *vanT*. VanT is a putative Serine racemase (mutations in Bc3-5) encoded within a VanG-type cluster [17] that was previously implicated in decreased vancomycin susceptibility in *C. difficile* [19] (S2 Data). *CD3437* encodes a predicted two-component system histidine kinase (mutations in Bc1 and 2), with its cognate response regulator encoded by *CD3438*. These genes had not previously been implicated in vancomycin resistance; however, the nearby *CD3439* encodes a putative D,D-carboxypeptidase that likely plays a role in modification of peptidoglycan through removal of the stem peptide terminal D-Ala. Based on these predicted functions, we propose renaming these genes *dacS* (*CD3437*, histidine kinase), *dacR* (*CD3438*, response regulator), and *dacJ* (*CD3439*, D,D-carboxypeptidase). Consistent with mutations at these loci playing key roles in vancomycin resistance, all 5 hyper-mutator replicate lines had mutations in the *vanG* operon (1 had a nonsynonymous mutation in *vanT* and 1 in *vanS*) or *dacS*, *dacR* (encoding the cognate response regulator) and *dacJ*. Interestingly, 1 strain (Bc8) had mutations in both *dacS* and *CD1523* (*vanS*), encoding a two-component system sensor histidine kinase that is thought to regulate the *vanG* operon in response to vancomycin, suggesting that the 2 pathways to resistance are not entirely mutually exclusive. *comR* encodes a homologue of the RNA degradosome component PNPase, suggesting that RNA stability may play a role in the *vanT*-associated mechanism of vancomycin resistance. Consistent with this possibility, in the third clone carrying a *vanT* mutation we observed coexisting mutations affecting *maa*, encoding a putative maltose O-acetyltransferase, and a 75-bp deletion that completely removed an intergenic region downstream of *rpmH* and before *rnpA*, encoding another predicted component of the RNA degradosome. By contrast, in the 2 clones carrying mutations in *dacS* we did not observe any mutations likely to affect RNA stability: one clone had no additional unique mutations, while the other had nonsynonymous mutations in *bclA3* and *CD3124*. *bclA3* encodes a spore surface protein with no known function in vegetative cells, while *CD3124* encodes an orphan histidine kinase of unknown function. Interestingly, three of the hyper-mutating lineages also had mutations in *CD3124*, an identical frameshift mutation in all three.

## Vancomycin-resistant clones display reduced fitness

To assess the wider consequences of evolved vancomycin resistance for bacterial phenotype, endpoint clones were phenotypically characterised for growth in vitro, sporulation efficiency,

and cell morphology. All 10 strains displayed significantly impaired growth in rich liquid media (S1 Fig), with particularly severe defects apparent for Bc10 and 11. Several strains were also impaired in sporulation (S2 Fig), with a wide range of phenotypes apparent, from a mild defect for Bc2 and 4 and delayed sporulation for Bc8, to more severe defects for Bc7, Bc9, and 10 and complete loss of sporulation for Bc11. These growth and sporulation defects were also accompanied by changes in cell length relative to the parental wild-type strain (S3 Fig). Principal component analysis (PCA) of the 5 wild-type-derived endpoint evolved clones (Fig 1D) revealed all 5 resistant strains followed similar evolutionary trajectories away from the parental wild type, associated with lower sporulation efficiency and growth defects, albeit with divergence among replicate lines in the extent of defects and cell size. However, there was no apparent subclustering by resistance mechanism.

## Population dynamics in evolving populations

Taken together, the phenotypic and genome sequence data for endpoint clones suggests that there are 2 alternative mechanisms of vancomycin resistance, one involving *dacS* and the other involving *vanT*. To better understand how selection acted upon these mechanisms, we next performed pooled population sequencing at passage 10, 20, and 30 to track mutation frequencies over time (Figs 2, S4, and S5). In total, discounting variants found in Bc1 P20, we identified 535 unique variants across the 10 parallel populations and 3 time points. We have removed Bc1 P20 from this analysis as the additional 520 variants identified in that sample alone likely reflect random mutation due to the emergence of a spontaneous hyper-mutator phenotype. Impacted genes clustered within 17 distinct functional classes by KEGG analysis (S6 Fig), with two-component systems and ABC transporters being particularly well represented. Focusing on the 2 main routes to resistance identified in endpoint clones, these data revealed highly contrasting selection dynamics, particularly in our wild-type replicate populations: mutations in *dacS* rapidly rose to high frequency, reaching fixation by P10. By contrast, mutations in *vanT* arose later and only reached fixation by P20 or 30, and were preceded by mutations at other sites which reached high frequency by P10 but that ultimately did not survive, being replaced by *vanT* mutants presumably conferring higher levels of vancomycin resistance. The 2 preceding high-frequency mutations in Bc3 (both T>TA) were very close together, separated by only 7 bp in an intergenic region downstream of *CD0482*, encoding a uridine kinase, and approximately 250 bp upstream of *glsA*, encoding a glutaminase. These mutations are outside of the likely promoter region [23] but it is possible that they affect regulation of *glsA* through alteration of the binding site for a regulatory protein or RNA. Interestingly, changes in glutamine metabolism have previously been linked to vancomycin resistance in *Staphylococcus aureus* [24]. The single high-frequency mutation in Bc5 at P10 is a nonsynonymous substitution in *CD3034*, introducing a Gly255Asp mutation in the encoded D-hydantoinase that may play a role in the synthesis of D-amino acids. Taken together, these data suggest that *vanT* is not required for first-step resistance; *vanT* mutations either provide higher-level vancomycin resistance, allowing them to supplant earlier mutations, or they require potentiating mutations to arise and be selected first. However, no consistent secondary mutations common to all populations with *vanT* mutations were identified.

## Recapitulation of *dacS* mutations confirms role in resistance

As the *dacJRS* cluster had not previously been implicated in vancomycin resistance, we validated the role of DacS in vancomycin resistance by recapitulating individual mutations in a clean genetic background. We chose the variant *dacS* allele identified in Bc1 (714G>T) and

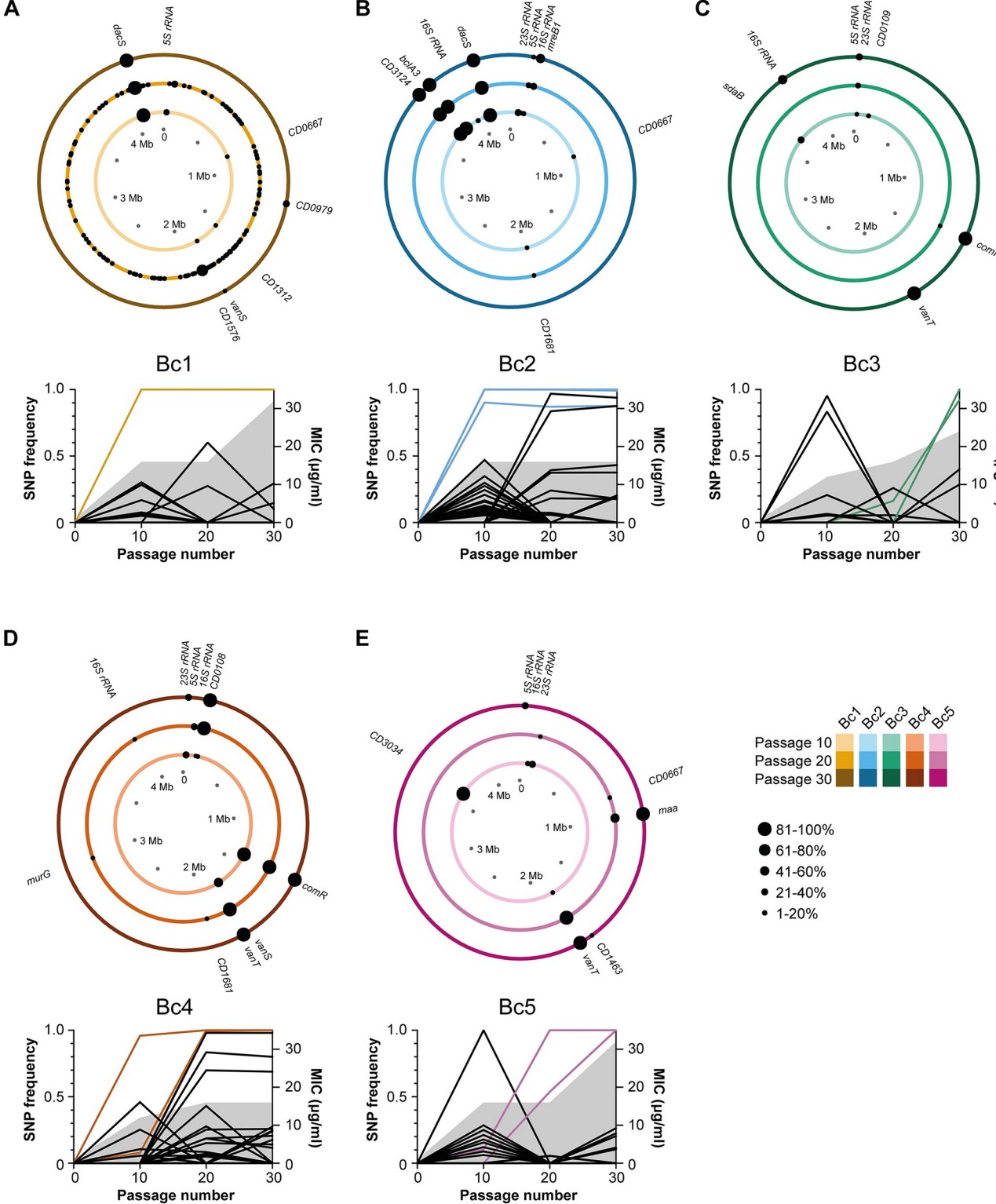

**Fig 2. Genomic location of gene variants over time.** Accumulation of variants in the wild-type *C. difficile* lineages Bc1 (**A**), Bc2 (**B**), Bc3 (**C**), Bc4 (**D**), and Bc5 (**E**). Each circle plot represents the 4.2 Mb genome of a single evolving population after 10 (inner ring), 20 (middle ring), and 30 passages (outer ring), with the locations of non-synonymous within gene variants indicated with black circles and the penetrance of each mutation in the population indicated by the size of the circle. The line graphs show the frequency of all variants (intergenic, synonymous, non-synonymous, frameshifts and nonsense) in each population. The vancomycin MIC for each population is also indicated by the shaded region. Mutations also identified in the

respective end point clone (Fig 1C) are highlighted by the coloured lines. Note population Bc1 evolved an apparent hyper-mutator phenotype prior to P20, with 520 variants identified at that time point. For simplicity, only variants present in P10 and P30 are labelled. The data underlying this figure, including a full list of all variants shown here and those in Bc1 P20, can be found in S3 Data. MIC, minimum inhibitory concentration.

the variant that evolved in parallel in Bc8 and Bc9 (548T>C). Recapitulated strains carrying only the *dacS* mutation of interest were constructed in the parental R20291Δ*PaLoc* by allelic exchange. Introduction of either mutation alone resulted in a 4-fold increase in the vancomycin MIC compared to the parental strain, confirming that DacS is playing a significant role in the evolved resistance we observed (Fig 3A). Interestingly, both mutations resulted in statistically significant growth defects (S7 and S9 Figs), suggesting they were responsible, at least in part, for the growth defects observed in the Bc1, Bc8, and Bc9 endpoint evolved clones. Neither mutation affected sporulation efficiency (S8 and S9 Figs). AlphaFold prediction of the DacS structure yields a plausible dimer model (Fig 3B) that is highly similar to previously characterised histidine kinases [25]. The Bc1 and Bc8/9 mutations described here both result in amino acid changes in the DacS cytoplasmic domain: Bc1 Glu238Asp within the predicted catalytic ATPase (CA) domain and Bc8/9 Val183Ala within the dimerization and histidine phosphorylation (DHp) domain. The impact of these mutations on the function of DacS was not clear but we hypothesised that DacS, along with its cognate response regulator DacR, could be regulating the expression of the D,D-carboxypeptidase encoded by *dacJ*. To examine this possibility, we extracted RNA from R20291Δ*PaLoc*, R20291Δ*PaLoc dacS*c.714G>T, and R20291Δ*PaLoc dacS*c.548T>C, both in the absence and presence of 0.5 μg/ml vancomycin, and assessed the expression of *dacS*, *dacR*, and *dacJ* by qRT-PCR (Fig 3C). Either point mutation resulted in a substantial up-regulation of expression of all 3 genes, varying from 4.8-fold increased transcription of *dacS* in R20291Δ*PaLoc dacS*c.714G>T to 94.6-fold increased transcription of *dacJ* in R20291Δ*PaLoc dacS*c.548T>C. Interestingly, these effects were independent of vancomycin suggesting that the two-component system responds to an as yet unknown signal. The genomic organisation of the *dacJRS* region (Fig 3D) and previous global transcription site mapping [23] suggests that *dacS* and *dacR* are transcribed from a single promoter upstream of *dacR* and that *dacJ* is transcribed from its own separate promoter. Our data demonstrated that both of these promoters are subject to regulation by the DacS/DacR two-component system, but could not differentiate between activation of a TCS that positively regulates these promoters or deactivation of a repressor. However, as complete deletion of *dacRS* had no effect on *dacJ* expression (Fig 3C), we conclude that the DacRS two-component system is a positive regulator of *dacJ*, and likely also *dacRS*, and that the mutations identified in Bc1, Bc8, and Bc9 result in increased TCS activity.

We hypothesised that the overexpression of DacJ, and the resulting removal of terminal D-Ala residues from peptidoglycan precursors, depleted vancomycin binding sites within the cell wall resulting in the observed reduction in vancomycin susceptibility in both strains with mutated *dacS*. To confirm the loss of accessible D-Ala-D-Ala in the cell walls of resistant strains, we incubated cells with vancomycin-BODIPY and visualised binding by fluorescence microscopy (Fig 4). The parental R20291Δ*PaLoc* strain displayed clear mid-cell vancomycin labelling in 34% of cells, consistent with the growth phase-dependent presence of pentapeptide precursors at the site of maximal peptidoglycan synthesis [19,29,30]. By contrast, *dacJ* overexpression in R20291Δ*PaLoc dacS*c.714G>T or *dacS*c.548T>C was associated with complete loss of vancomycin labelling. To confirm that this difference was due to DacJ, we constructed a *dacJ* deletion in resistant endpoint isolate Bc1, which partially restored vancomycin labelling and reduced the vancomycin MIC 8-fold (Fig 4 and Table 1).

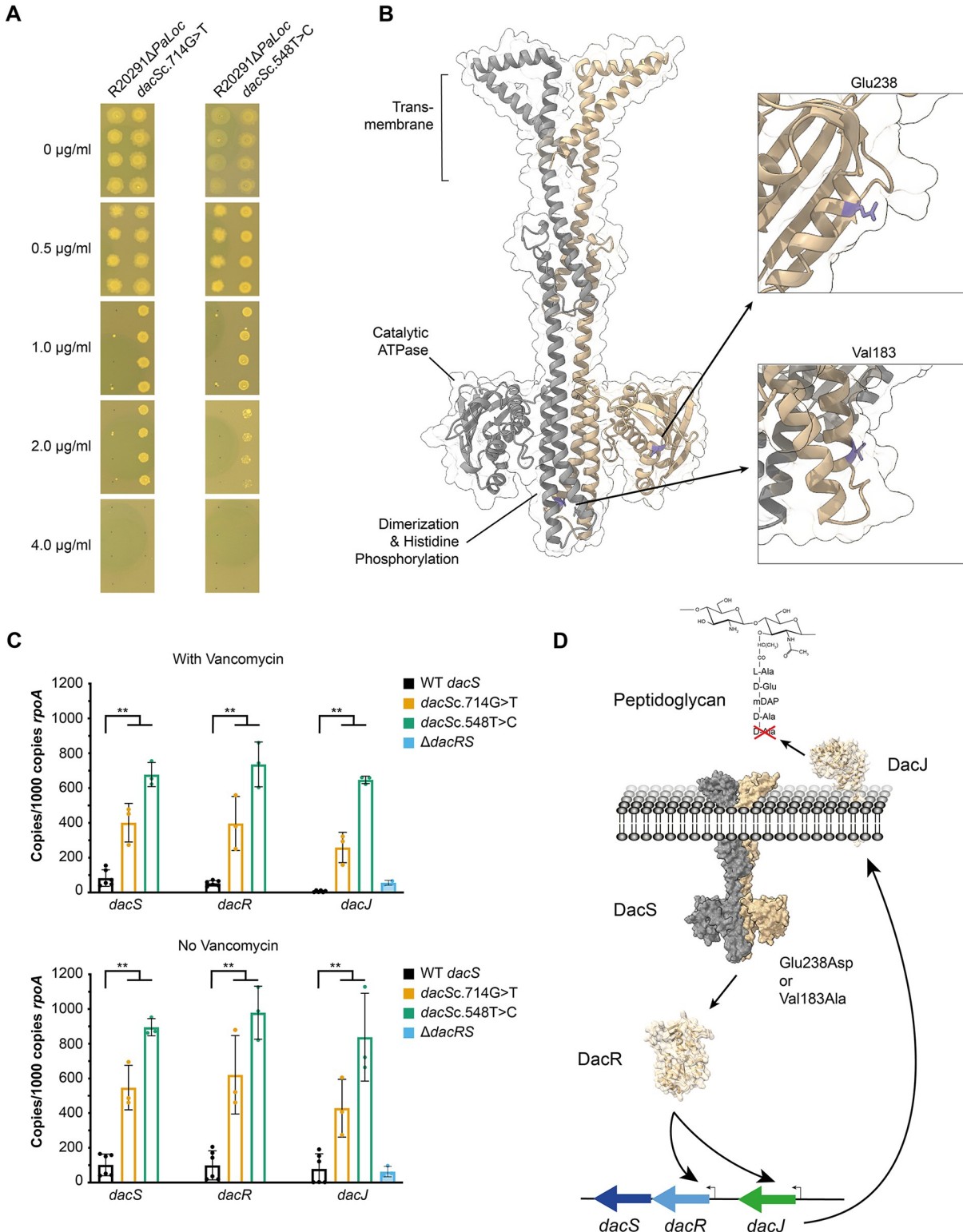

**Fig 3. *dacS* mutations lead to dysregulation of *dacJRS*.** (**A**) Vancomycin MICs of R20291Δ*PaLoc*, R20291Δ*PaLoc dacS*c.714G>T, and R20291Δ*PaLoc dacS*c.548T>C as determined by agar dilution. Assays were performed in biological triplicate and technical duplicate, 4 representative spots are shown for each strain. (**B**) AlphaFold model of DacS as a dimer [26]. The transmembrane domains were identified using DeepTMHMM [27] and the Catalytic ATPase and Dimerization and Histidine Phosphorylation domains were predicted using InterProScan [28]. The locations of Val183 and Glu238 are highlighted in purple on one chain. (**C**) qRT-PCR analysis of *dacJRS* expression in

R20291Δ*PaLoc* (black bars), R20291Δ*PaLoc dacS*c.714G>T (yellow bars), R20291Δ*PaLoc dacS*c.548T>C (green bars), and R20291Δ*PaLoc*Δ*dacRS* (blue bars). Expression was quantified against a standard curve and normalised relative to the house-keeping gene *rpoA*. Assays were performed in biological and technical triplicate. Statistical significance was calculated using a two-way ANOVA with the Tukey–Kramer test, ** = $P < 0.001$. The data underlying this panel can be found in S5 Data. (**D**) Lack of *dacJ* up-regulation in R20291Δ*PaLoc*Δ*dacRS* suggests that phosphorylated-DacR acts as an activator of the 2 promoters in the *dacJRS* cluster, with DacS Glu238Asp or Val183Ala substitutions constitutively activating the function of the TCS, respectively. The consequence is over-expression of DacJ which is then translocated to the cell surface where it can cleave the terminal D-Ala residue in nascent peptidoglycan, thereby preventing vancomycin binding. MIC, minimum inhibitory concentration.

## Vancomycin resistance in Bc1 is multifactorial

Recapitulation of the *dacS*c.714G>T mutation, originally identified in the genome sequenced resistant endpoint clone from replicate Bc1, resulted in only a 4-fold increase in vancomycin MIC in the recapitulated mutant, compared with the 16–32-fold increase seen for the evolved clone. Because *dacS*c.714G>T was the only mutation we observed in the evolved clone, this discrepancy in MICs suggested that the evolved Bc1 clone had additional mutations that were not detected by our short-read Illumina-based pipeline. To address this, we re-sequenced end-point isolates Bc1-5 using a long-read technology (Oxford Nanopore) and combined this with the existing short-read data to generate high-quality closed genomes. No additional mutations were identified in isolates Bc3-5 (S4 Data). However, previously unobserved additional mutations were identified in the evolved clones from replicates Bc1 and Bc2. Two additional single-nucleotide insertions were identified in evolved clone Bc2, causing frameshifts in *CD0794* and *CD1871*, encoding a conserved hypothetical protein and an ABC transporter permease, respectively. Two additional large indels were identified in Bc1, a 44-bp deletion downstream of *CD0978* and upstream of *CD0980* (1,197,357_1,197,400del) and a 30-bp duplication within the ORF of *vanS* (*vanS*c.367_396dup), encoding the *vanG* cluster histidine kinase. In the annotated R20291 genome, there is a very small 168-bp ORF upstream of *CD0980*, annotated *CD0979*. However, as other *C. difficile* genome sequences, including that of the well-characterised strain 630 [32] do not show an equivalent ORF in this region and comprehensive RNAseq analysis [23] did not detect any transcription associated with *CD0979*, we conclude that this is likely a misannotation and that the observed deletion affects an intergenic region. To assess the contribution of these large indels to vancomycin resistance in Bc1, we constructed a series of R20291Δ*PaLoc* mutant strains with *dacS*c.714G>T, 1,197,357_1,197,400del and *vanS*c.367_396dup in all possible combinations of single, double, and triple mutations. The *vanS*/*dacS* double mutant fully recapitulated the increased vancomycin resistance seen in the evolved Bc1 clone (16 μg/ml, Table 1), whereas the *vanS* and *dacS* single mutants only displayed a 2- and 4-fold increase in MIC, respectively, suggesting that these mutations interacted synergistically with respect to the resulting vancomycin resistance level. The 1,197,357_1,197,400del mutation made no apparent contribution to vancomycin resistance in any of the constructed mutants.

Analysis of growth profiles of all recapitulated single, double, and triple mutants revealed severe growth defects similar to that seen in the evolved Bc1 clone for all recapitulated strains that included the *dacS* point mutation, suggesting that this mutation alone was largely responsible for the growth defect (S7 Fig). Sporulation of the triple mutant was very similar to both the evolved Bc1 clone and the parental R20291Δ*PaLoc* (S8 Fig).

Mutations in *vanS* have previously been implicated in reduced vancomycin susceptibility through increased expression of the enzyme-encoding genes in the *vanG* cluster. To assess if a similar mechanism was at play in Bc1, we performed qPCR on all of the genes in the *vanG* cluster in R20291Δ*PaLoc vanS*c.367_396dup (Fig 5A). The *vanS* mutation had no effect on the expression of *vanR* in either the presence or absence of 0.5 μg/ml vancomycin but,

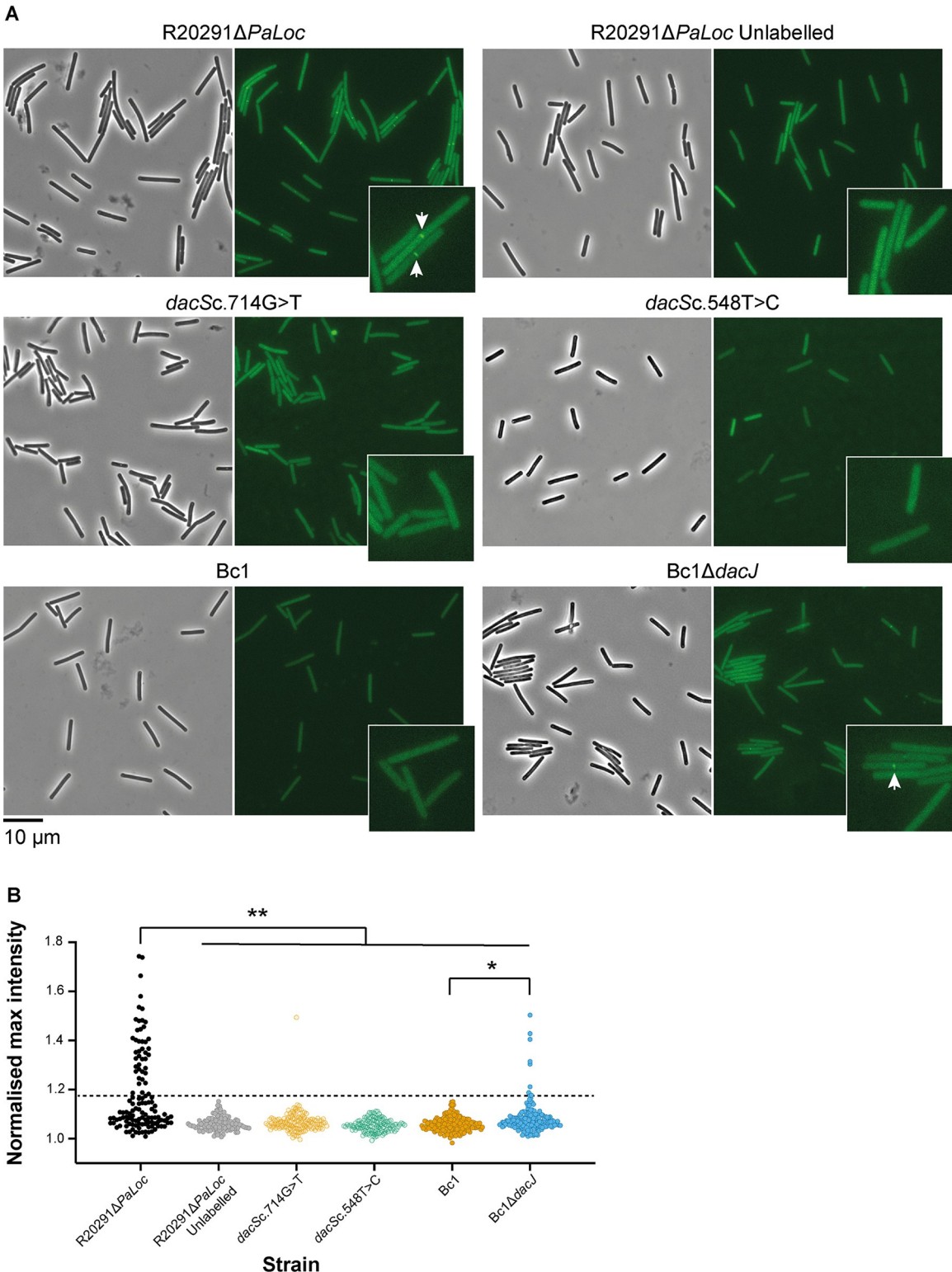

**Fig 4. DacJ activity depletes vancomycin binding sites in the cell wall.** (**A**) Representative phase contrast (left) and fluorescence (right) images of R20291Δ*PaLoc*, R20291Δ*PaLoc dacS*c.714G>T, R20291Δ*PaLoc dacS*c.548T>C, Bc1 and Bc1Δ*dacJ* labelled with vancomycin-BODIPY. A control of unlabelled R20291Δ*PaLoc* (top right) is included to show the background fluorescence that *C. difficile* exhibits in the presence of oxygen [31]. Vancomycin labelling of accessible D-Ala-D-Ala, present during active synthesis of peptidoglycan, is indicated with white arrows. (**B**) The normalised maximum fluorescent intensities at the centre of individual cells showed that 34% of

R20291ΔPaLoc cells had bound vancomycin, compared to no cells in the unlabelled control of the same strain. R20291Δ*PaLoc dacS*c.714G>T and R20291Δ*PaLoc dacS*c.548T>C and the evolved endpoint clone Bc1, which all overexpress *dacJ*, were not labelled when incubated with vancomycin-BODIPY. However, when *dacJ* was deleted in Bc1 partial labelling was restored (4% of cells were labelled at their mid-point). This was consistent with the incomplete re-sensitisation to vancomycin also observed with this mutant. The black dashed line indicates the normalised max intensity threshold used to delineate cells that have detectable levels of vancomycin-BODIPY at their midplane from those without (obtained by direct visual inspection, normalised max intensity = 1.18). Asterisks indicate the *P*-values from pairwise comparisons using a two-tailed Fisher's exact test; ** = *P* < 0.0001, * = *P* < 0.05. The data underlying panel **B** can be found in S5 Data.

surprisingly, a significant decrease was observed in the *vanS* transcript copy number in both conditions (4.5- to 4.8-fold decrease). As these 2 genes are thought to be transcribed in a single bicistronic mRNA [23], this observation suggests that the duplication within the *vanS* ORF either impacts the stability of the 3′ end of the transcript or transcriptional efficiency past *vanR*. Despite the apparent decrease in the *vanS* transcript copy number, the expression of *vanG*, *vanXY*, and *vanT* were all significantly increased in the *vanS* mutant (from a 10-fold increase in transcription of *vanG* in the presence of vancomycin to a 136-fold increase in *vanG* without vancomycin).

## Discussion

Vancomycin is one of the few antibiotics in routine use for treatment of CDI worldwide and is now the frontline drug of choice in the UK [9]. High-level vancomycin resistance is widespread in *Enterococcus* spp. and in *S. aureus* but has yet to be reported in *C. difficile*, where there are few verifiable reports of reduced susceptibility in clinical strains, despite anecdotal reports of vancomycin treatment failure [33]. However, it is not clear if this apparent lack of resistance reflects an underlying constraint upon the emergence of resistance in this species or is simply an artefact of a lack of routine monitoring in the clinic. Here, we show, using laboratory experimental evolution, that *C. difficile* can rapidly evolve high-level vancomycin resistance via 2 alternative mechanisms, but that increased resistance is associated with severe pleiotropic effects, including growth and sporulation defects, which may act to limit the emergence of resistance in clinical settings.

Under ramping vancomycin selection, resistance emerged rapidly in all 10 replicate lines reaching 16- to 32-fold higher MIC within approximately 250 generations. Whole genome sequencing of individual clones from each population at the end of the evolution revealed 2 evolutionary pathways to resistance, centring around mutations in *vanT*, encoding the Serine/

**Table 1. Vancomycin MICs of strains recapitulating Bc1 mutations.**

| Strain | MIC (µg/ml) |
| --- | --- |
| R20291Δ*PaLoc* | 1 |
| Bc1 | 16 |
| Bc1Δ*dacJ* | 2 |
| R20291Δ*PaLoc dacS*c.714G>T | 4 |
| R20291Δ*PaLoc* 1,197,357_1,197,400del | 1 |
| R20291Δ*PaLoc vanS*c.367_396dup | 2 |
| R20291Δ*PaLoc dacS*c.714G>T *vanS*c.367_396dup | 16 |
| R20291Δ*PaLoc dacS*c.714G>T 1,197,357_1,197,400del | 4 |
| R20291Δ*PaLoc* 1,197,357_1,197,400del *vanS*c.367_396dup | 2 |
| R20291Δ*PaLoc dacS*c.714G>T *vanS*c.367_396dup 1,197,357_1,197,400del | 16 |

MIC, minimum inhibitory concentration.

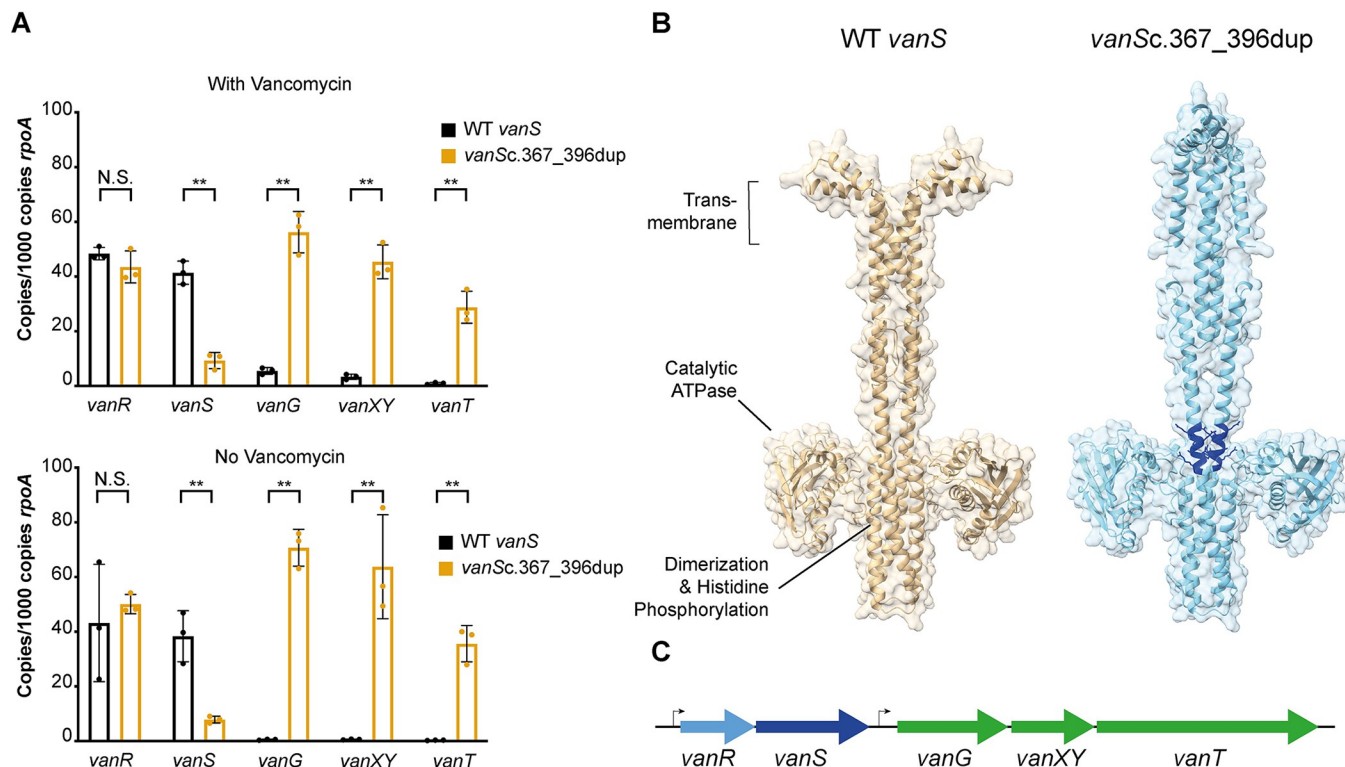

**Fig 5. *vanS* mutations lead to dysregulation of *the vanG* cluster.** (**A**) qRT-PCR analysis of expression of the genes in the *vanG* cluster in R20291Δ*PaLoc* (black bars) and R20291Δ*PaLoc vanS*c.367_396dup (orange bars). Expression was quantified against a standard curve and normalised relative to the house-keeping gene *rpoA*. Assays were performed in biological and technical triplicate. Statistical significance was calculated using a two-way ANOVA with the Tukey–Kramer test, ** = $P < 0.001$. (**B**) AlphaFold model of VanS as a dimer with (right) and without (left) the *vanS*c.367_396dup encoded 10 amino acid duplication [26]. The transmembrane domains were identified using DeepTMHMM [27] and the Catalytic ATPase and Dimerization and Histidine Phosphorylation domains were predicted using InterProScan [28]. The location of the 10 amino acid duplication is highlighted on both chains in dark blue. The insertion results in an approx. 90° rotation of the transmembrane domains relative to the catalytic ATPases. (**C**) Genomic organisation of the *vanG* cluster. The data underlying this panel **A** can be found in S5 Data.

Alanine racemase component of a VanG-type cluster, and mutations in a cluster of genes encoding a two-component system and a putative D,D-carboxypeptidase (*dacJRS*). The VanG cluster is common in *C. difficile* strains but its potential role in vancomycin resistance had been disputed [17,18]. More recently however, mutations in the VanRS two-component system that derepress the rest of the cluster and reduce vancomycin susceptibility have been identified in both laboratory evolution experiments and in clinical isolates [19,20]. These observations confirm that changes to the expression of genes in the VanG cluster can indeed contribute to reduced vancomycin susceptibility. Interestingly, we detected *vanS* SNPs only transiently in 2 replicate lines (Bc4 and 11), suggesting they had only a limited contribution to resistance or were accompanied by severe fitness defects in this genetic background (Fig 4). A further *vanS* mutation, a larger 30-bp duplication, was observed in endpoint isolate Bc1, resulted in significant overexpression of the whole *vanG* cluster and a small increase in the vancomycin MIC, similar to *vanS* mutations that were described previously [19]. In contrast, mutations to *vanT* were common, fixing in 4 of 10 replicate lines. Unfortunately, we have been unsuccessful in recapitulating *vanT*-mediated resistance, despite introducing all 4 of the identified *vanT* mutations individually into the parental R20291Δ*PaLoc* strain. Indeed, recapitulation of all SNPs and indels identified in the Bc3 endpoint clone in R20291Δ*PaLoc* resulted in only a modest 2-fold increase in vancomycin MIC. Given the phenotypic differences between

apparently genetically identical strains, we hypothesise that currently unknown epigenetic or regulatory effects may play an important role in this mechanism of resistance. Mutations in the *dacJRS* cluster were also extremely commonly observed and often rose to high frequency, with variant *dacS* alleles fixing in 3 populations (Bc1, 2, and 9), an identical *dacR*c.532A>G variant fixing in 2 populations (Bc7 and 10) and a *dacJ* variant fixing in Bc9. A mutation in *dacS* also transiently fixed in Bc8 at P20 before being subsequently outcompeted. Interestingly, the *dacS* mutation identified in Bc8 at P20 (548T>C) was identical to that observed in endpoint isolates from Bc9. None of these genes had been previously linked to vancomycin resistance but the predicted D,D-carboxypeptidase activity of DacJ points to a plausible mechanism through removal of the terminal D-Ala residue in nascent peptidoglycan [34,35] (Fig 3D). We hypothesised that the two-component system encoded by *dacS* and *dacR* was regulating expression of *dacJ*, providing a potential mechanistic link to vancomycin resistance for all of these mutations. This was confirmed by recapitulation of 2 distinct *dacS* mutations (from Bc1 and Bc8/9) in a clean genetic background, leading to overexpression of both *dacJ* and the *dacSR* bicistronic operon. Complete deletion of *dacRS* had no effect on *dacJ* expression suggesting that the *dacS* SNPs observed in Bc1, Bc8, and Bc9 endpoint clones are gain of function mutations that increase activity of the two-component system. Importantly, this effect was independent of vancomycin, leaving open the possibility that the wild-type *dacJRS* could still contribute to vancomycin resistance in the presence of the appropriate activating signal. Overexpression of *dacJ* due to SNPs in *dacS* led to a significant reduction in accessible vancomycin binding sites at the mid-cell but, on its own, this only resulted in a modest increase in vancomycin MIC. To fully recapitulate resistance, we had to combine the Bc1 *dacS* SNP (*dacS*c.714G>T) with the *vanS* duplication (*vanS*c.367_396dup). Interestingly, the contribution of these 2 mutations was synergistic rather than simply additive, pointing to a degree of functional collaboration between the 2 pathways. Interestingly, a recent publication demonstrated that DacS (CD630_35990) could induce its own expression and that of the *vanG* cluster genes in response to vancomycin in the lab-adapted strain 630Δerm [36]. However, we have shown here that DacS in strain R20291 does not respond to vancomycin (Fig 3C). These differences in behaviour are likely due to sequence variation in the extracellular sensor domain of DacS; R20291 DacS L28 is substituted for Ser in 630, E34 for Gly, and F48 for Leu.

Analysis of population genetic dynamics over the course of the evolution also revealed intriguing differences in the timing of *dacJRS* vs *vanT* mutations (summarised in Fig 6). *dacJRS* mutations had fixed by P10 in nearly every population in which they persisted to the

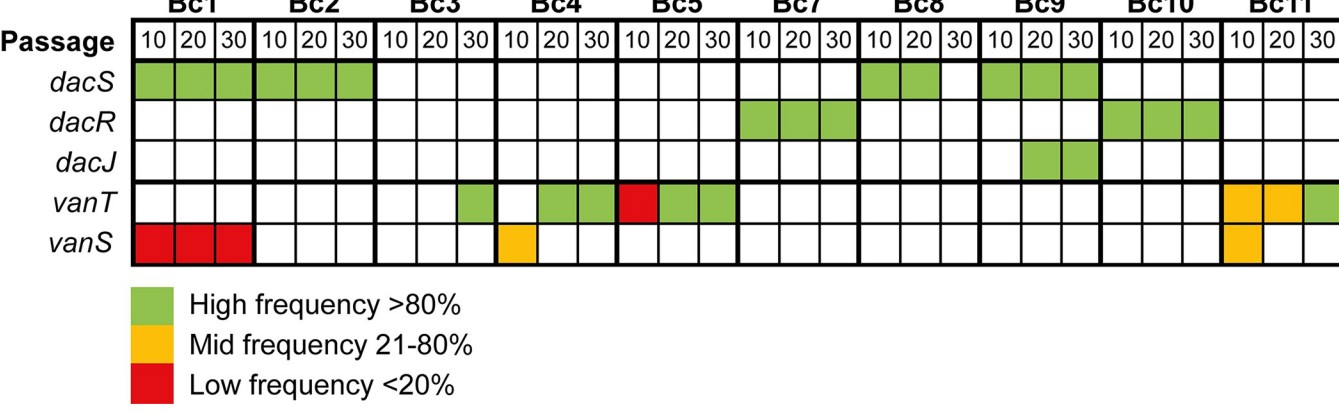

**Fig 6. Two distinct pathways to resistance.** Relative frequencies and time of emergence of mutations in genes *dacS*, *dacR*, *dacJ*, *vanT*, and *vanS* across all 10 evolving populations. The data underlying this figure can be found in S3 Data.

end of the evolution, the exception being Bc9 *dacJ* which first appeared and fixed at P20. However, that population also had a preceding *dacS* mutation that had fixed by P10. By contrast, mutations in *vanT* typically did not fix until P20 or P30, although were sometimes present at lower frequency at earlier time points. Delayed emergence of *vanT* variants may also indicate a reliance on preexisting potentiating mutations, although no consistent secondary mutations were found in all *vanT* lineages. The 2 resistance mechanisms also seemed to be mutually exclusive, *vanT* mutations did not co-occur with *dacJRS* in any population or endpoint isolate. It is possible that early emergence of mutations in *dacJRS* precludes subsequent mutations to *vanT* and commits the population to the *dacJRS* pathway.

On average, the *mutSL* hyper-mutating populations became vancomycin resistant more quickly than the equivalent wild types and this was associated with a dramatically increased accumulation of SNPs throughout the evolution at the population level, and also apparent in the endpoint evolved clones. Evidence from both population and endpoint sequencing suggests that, at least in part, this reflects the hyper-mutators exploring a wider array of possible beneficial mutations. However, the wide variation in both the numbers and nature of accompanying mutations suggests that many of these may also be random hitchhiking mutations.

In all populations, emergence of resistance in our experiment was accompanied by fitness costs and pleiotropic phenotypic effect, albeit of varying severity between populations. Importantly, severe sporulation defects were associated with increased vancomycin resistance in 2 lineages. As spores are the infectious form of *C. difficile* and an absolute requirement for patient to patient transmission [37], these sporulation defects would likely impair the infectivity and onward transmission of such resistant genotypes. The precise mechanistic cause of the sporulation defects in these evolved clones is unclear, in part because sporulation is a complex and poorly understood cell differentiation process [38,39], but if a similar trait were to emerge in a patient during vancomycin treatment this would be an evolutionary dead end. The fitness costs and deleterious pleiotropic consequences of vancomycin resistance mutations may well explain the delayed emergence of vancomycin resistance in CDIs in the clinic. However, it is also possible that the accumulation of subsequent compensatory mutations could mitigate the immediate fitness costs of resistance, as has been seen in long-term evolution experiments [40,41]. Ramping antibiotic selection drove recurrent displacement of resistance mutations in several evolving populations, with early moderately resistant variants being supplanted by distinct genetic lineages with higher-level resistance, highlighting a diversity of evolutionary pathways to vancomycin resistance available to *C. difficile*. Given the increasing reliance on vancomycin in the treatment of CDI, it is crucial that we better understand these multiple genetic routes to resistance. The question of how likely *C. difficile* vancomycin resistance evolution is in the real world remains open but this work, and other efforts towards understanding possible mechanisms of resistance, will hopefully provide a roadmap to guide genomic surveillance efforts.

## Methods

### Strains and growth conditions

All strains used or generated in the course of this study are described in S1 Table. *C. difficile* was routinely cultured on brain heart infusion (BHI) agar and in tryptone yeast (TY) broth. *C. difficile* was grown in an anaerobic cabinet (Don Whitley Scientific) at 37°C, with an atmosphere composed of 80% $N_2$, 10% $CO_2$, and 10% $H_2$. Media was supplemented with thiamphenicol (15 μg/ml) (Sigma) and colistin (50 μg/ml) (Sigma) as appropriate. For counter selection against plasmids bearing the *codA* gene, *C. difficile* defined media with 5-fluorocytosine (CDDM 5-FC) was used as described previously [42]. *E. coli* was routinely cultured in Luria–

Bertani (LB) broth or agar at 37°C and supplemented with chloramphenicol (15 μg/ml) (Acros Organics) or kanamycin (50 μg/ml) (Sigma) as appropriate.

## Molecular biology

All oligonucleotides and plasmids are described in S2 and S3 Tables, respectively. Plasmid miniprep, PCR purification, and gel extractions were performed using GeneJET kits (Thermo Fisher). High-fidelity PCR amplification was performed using Phusion polymerase (NEB), and standard PCR amplification was performed using Taq mix red (PCRBIO), according to manufacturers' instructions. Gibson assembly primers were designed using NEBBuilder (NEB) and restriction digestion and DNA ligation was performed using enzymes supplied by NEB. Plasmids were transformed into NEB5a (NEB) or CA434 competent *E. coli* cells according to the NEB High Efficiency Transformation Protocol. The sequences of cloned fragments were confirmed using Sanger sequencing (Genewiz, Azenta Life Sciences, Germany).

## *C. difficile* mutagenesis

*C. difficile* strain R20291 was modified for use in evolution experiments and to recapitulate individual mutations. Homology arms for introducing mutations onto the *C. difficile* chromosome by allelic exchange were generated either by Gibson assembly of PCR products or synthesised by Genewiz (Azenta Life Sciences) and then subsequently cloned between BamHI and SacI sites in pJAK112 [23], a derivative of pMTL-SC7215 [42]. Following confirmation by Sanger sequencing, plasmids were transformed into *E. coli* CA434 and transferred to *C. difficile* by conjugation [43]. Homologous recombination was performed as previously described [42] and mutations were confirmed by PCR and Sanger sequencing of mutated regions.

R20291 was rendered avirulent via deletion of the complete pathogenicity locus (encoding toxins A and B) using plasmid pJAK143 [44], yielding strain R20291Δ*PaLoc*. This strain was then further modified by deletion of DNA repair operon *mutSL* to create a hyper-mutator strain; 1.2 kb up and downstream of the *mutSL* operon was amplified by PCR using oligonucleotides RF2066 and RF2067, and RF2068 and RF2069 (S2 Table), respectively, and cloned between BamHI and SacI sites in plasmid pJAK112. The resulting plasmid, pJEB002, was then conjugated into R20291Δ*PaLoc* and allelic exchange performed as described above, yielding R20291Δ*PaLoc*Δ*mutSL*. Enumeration of colony-forming units (CFUs) on BHI agar containing rifampicin (0.015 μg/ml) suggested this strain has a mutation rate approximately 20-fold higher than R20291Δ*PaLoc*. Five barcoded variants of both R20291Δ*PaLoc* and R20291Δ*PaLoc*Δ*mutSL* were then generated by insertion of unique sequencing barcodes. Briefly, pJAK081 (identical to pJAK080 [23] but in the pMTL-SC7215 backbone) was modified by introduction of a synthetic DNA fragment containing a multiple cloning site and a 9-bp sequencing barcode (Barcode 1), flanked by the *fdx* and *slpA* terminators, between the existing homology arms for insertion of DNA between *CD0188* (*pyrE*) and *CD0189* in the R20291 genome. A second plasmid (pJAK202) containing Barcode 2 was constructed in the same manner and the rest (pJAK203-205 and 207–211) were generated via site-directed mutagenesis using pJAK201 as a template. These 10 plasmids were then used to generate R20291Δ*PaLoc* Bc1-5 and R20291Δ*PaLoc*Δ*mutSL* Bc7-11. Recapitulated strains were generated by introducing deletion, insertion, or point mutations into R20291Δ*PaLoc*. For each desired mutation, an approximately 2-kb synthetic DNA fragment, centred on the mutation of interest, was cloned between BamHI and SacI sites in pJAK112. The resulting plasmids (S3 Table) were conjugated into R20291Δ*PaLoc*, and allelic exchange was performed as described above.

## Evolution

Directed evolution of *C. difficile* was performed using a broth-based gradient approach in which 10 individually barcoded parallel lines were evolved for a period of 30 passages. A 6-well plate for each of the parallel lines was prepared for each passage using 4 ml of TY broth with vancomycin, spanning a gradient of 0.25 to 8× the current MIC, as determined from the most recent passage for each line, allowing the gradient to rise with increasing resistance.

The evolution was initiated using overnight cultures from single colonies, adjusted to $OD_{600nm}$ 0.1, and 10 μl was added to each well, before incubating for 48 h at 37°C. Plates were visually inspected after 48 h and 10 μl of the well with the highest antibiotic concentration supporting growth was used to inoculate the wells of the subsequent passage. For each parallel line, a control well was passaged without antibiotic; 1 ml of a population, and the corresponding control, was frozen at −80°C in 15% glycerol whenever the MIC increased, and after passages 10, 20, and 30.

## gDNA extraction

gDNA was obtained from *C. difficile* cultures using the phenol–chloroform method as described previously [38]. DNA concentration was quantified using Qubit, and purity was assessed via microvolume spectrometry.

## Sequencing

Endpoint (P30) isolates and respective controls were gained through plating P30 populations and culturing single colonies. Vancomycin MICs were determined by agar dilution (described below), and gDNA extraction and quantification were performed as above. Library prep (Nextera XT Library Prep Kit (Illumina, San Diego, United States of America)) and 30× illumina sequencing (NovaSeq 6000, 250-bp paired end protocol) of isolates was performed at MicrobesNG (Birmingham, UK). Reads were trimmed at MicrobesNG using Trimmomatic (v0.30) [45] with a sliding window quality cutoff of Q15.

WT-derived endpoint isolates, the same as those described above, were also sequenced by nanopore. Library prep (Oxford Nanopore Technologies SQK-RBK114.96 kit (ONT, UK)) and nanopore sequencing (GridION, FLO-MIN114 (R.10.4.1) flow cell) was performed at MicrobesNG (Birmingham, UK).

Pooled sequencing of the 10 evolved and 10 control populations was performed at 3 time points—P10, P20, and P30, and 1 ml of the well with highest antibiotic concentration supporting growth was taken after 48 h, harvested via centrifugation, and frozen at −20°C. gDNA extraction and quantification were performed as above. Library prep (Nextera DNA Flex Library Prep Kit (Illumina, San Diego, USA)) and 250× Illumina sequencing (NovaSeq 6000, 150-bp paired end protocol) was performed at SNPsaurus (Oregon, USA). Reads were trimmed using Trimmomatic (v0.39) with the following criteria: leading:3; trailing:3; slidingwindow:4:15; minlen:36.

## Sequencing analysis pipeline: Isolates

Trimmed reads were checked using FastQC (v0.11.9) [46] to ensure sufficient quality for analysis. Analysis was primarily performed using a custom pipeline, built based on a resistant mutant analysis pipeline [47]. Reads were aligned to the reference (*C. difficile* R20291, accession number: FN545816) using BWA-mem (v0.7.17) [48] and sorted using SAMtools (v1.43) [49]. Coverage across the genome was inferred using Bedtools (v2.30.0) [50] genomecov and map functions. PCR duplicates were removed via Picard (v2.25.2) (http://broadinstitute.

github.io/picard/). The mpileup utility within SAMtools (v1.43) was used to generate the mpileup file required for Varscan. Variants were then called using Varscan (v2.4.3–1) [21] mpileup2cns (calling SNPs and indels) using the following parameters: min-coverage 4; min-reads2 4; min-var-freq 0.80; *p*-value 0.05; variants 1; output-vcf 1. Here, a minimum of 4 reads were required to support a variant, and the cutoff for variant calling was 80%. Vcf files were annotated using snpEff (v5.0) [51]. Variants were also called using Breseq (v0.35.5) [22], using default parameters, and putative variants were retained if detected in both analysis pipelines. All variants were manually verified using IGV (v2.8.6) [52]. Variants that were also called in control lines were removed using Varscan (v2.4.3–1) compare, generating a list of variants unique to vancomycin resistant lines.

Nonsynonymous mutations occurring within genes were plotted using a previously published custom script in RStudio (v4.1.0) utilising the Plotrix package [53] to visualise parallel evolution.

### Sequencing analysis pipeline: Isolates (long-read)

Long-read genome assembly was performed at MicrobesNG (Birmingham, UK): Reads were assembled using Flye (v2.9.2-b1786) [54], polished using Medaka (v1.8.0) (https://github.com/nanoporetech/medaka), and annotated using Bakta (v1.8.1) [55]. For a given isolate, Nanopore and Illumina data were amalgamated using Polypolish (v0.6.0) [56]. Short read sequences were aligned to the Nanopore assembly using BWA-mem (v0.7.17) [48], using the -a parameter. Alignments were filtered by insert size using Polypolish with default parameters. Assemblies were polished using Polypolish and annotated with Prokka (v1.14.6) [57]. Variants were called using Snippy (v4.6.0) (https://github.com/tseemann/snippy) with default parameters. Identified variants were manually verified using IGV (v2.8.6) [52].

### Sequencing analysis pipeline: Populations

The 10 evolved and 10 control populations, sequenced at P10, P20, and P30, were evaluated using a population analysis methodology that followed the same custom pipeline as for isolate analysis, with modifications to SNP calling parameters and filtering. InSilicoSeq (v1.5.4) [58] was used to rapidly simulate realistic sequencing data at a range of coverage depths (80×, 100×, 300×) based on the R20291 genome, with SNPs seeded at 5% frequency. Simulated sequences were analysed using the custom pipeline, and a range of Varscan parameters (*P*-values, min-coverage, min-reads2) were trialled to generate a set of parameters to accurately call variants without false positives.

Varscan parameters used to call population variants were dependent on average coverage depth, allowing a strong evidence base while calling low-frequency variants. Samples were separated by average coverage: the following Varscan (v2.4.3–1) mpileup2cns parameters were used for samples with 100× or higher average coverage: min-coverage 80; min-var-freq 0.05; *P*-value 0.05; variants 1; output-vcf 1. A minimum of 4 reads were required to support a variant, and the cutoff for variant calling was 5%. For samples with average coverage below 100×, the following Varscan (v2.4.3–1) mpileup2cns parameters were used: min-coverage 4; min-reads2 4; min-var-freq 0.05; *p*-value 0.05; variants 1; output-vcf 1. As with high-coverage samples, a minimum of 4 reads were required to support a variant, meaning the minimum variant frequency which could be called was inversely scaled (4/coverage at position), with a minimum frequency of 5%. This process identified a list of variants within each line at each time point and their respective frequencies.

As in the isolate pipeline, population variants were filtered using Varscan (v2.4.3–1) compare, removing variants that were also present in control lines to generate a list of variants

unique to vancomycin-resistant lines. Further filtering of this dataset, again using Varscan (v2.4.3–1) compare, discarded variants that never reached above 10% frequency by the end of the evolution. Variants were also manually inspected to ensure no variants remained the same frequency across all time points.

Mutations occurring within genes (except those in Bc1 P20) were coloured according to their barcoded replicate line and displayed in KEGG Mapper–Color [59] to view KEGG pathways.

Nonsynonymous mutations occurring within genes and rRNAs were plotted using a previously published custom script in RStudio (v4.1.0) utilising the Plotrix package [53] to visualise population dynamics, where point size indicated mutation frequency in the population.

## Strain fitness analysis

Growth curves were performed anaerobically in 96-well plates using a Stratus microplate reader (Cerillo). Overnight cultures of *C. difficile* were diluted to $OD_{600nm}$ 0.05 and incubated at 37˚C for 1 h. Plate lids were treated with 0.05% Triton X-100 + 20% ethanol. Plates were prepared using 200 µl of equilibrated culture. Samples were measured at minimum in biological and technical triplicate. The $OD_{600nm}$ was measured every 3 min over a 24-h period. Data was plotted in Graph pad Prism (v9.0.2) and was analysed in RStudio (v4.1.0) using the Growth-Curver package (v0.3.0) [60].

To assess sporulation efficiency, triplicate overnight *C. difficile* cultures were adjusted to $OD_{600nm}$ 0.01 and grown for 8 h. Cultures were then adjusted again to $OD_{600nm}$ 0.01, subcultured 1:100 into 10-ml BHIS broth, and grown overnight to obtain early stationary phase spore-free cultures (T = 0). At T = 0, and the following 5 days, total viable counts were enumerated by spotting 10-fold dilutions in technical triplicate onto BHIS agar with 0.1% sodium taurocholate. Colonies were counted after 24-h incubation. Spore counts were enumerated using the above method following heat treatment (65˚C for 30 min).

To visualise cell morphology, *C. difficile* samples were harvested via centrifugation, washed twice in PBS, and fixed in 4% paraformaldehyde, before harvesting and resuspension in $dH_2O$. Samples were mounted in 80% glycerol and imaged using a 100× Phase Contrast objective on a Nikon Ti eclipse widefield imaging microscope using NIS elements software. Images were analysed in Fiji (v2.9.0) using MicrobeJ (v5.13l) [61].

## Vancomycin MICs

MICs were obtained via standard agar dilution methods [62]. Briefly, overnight cultures were adjusted to $OD_{600nm}$ 0.1, corresponding to between $1.06 \times 10^7$ and $1.83 \times 10^7$ CFUs for all strains analysed, and 2.5 µl samples were spotted in biological triplicate and technical duplicate onto BHI plates with ranging antibiotic concentrations. MICs were determined after 48 h incubation, and plates were imaged using a Scan 4000 colony counter (Interscience).

## Principal component analysis

WT (Bc1-5) P30-resistant isolates were characterised in terms of their sporulation efficiency, growth rate, MIC, and cell length. These were compared, along with the WT ancestor from the start of the evolution, in a PCA. The PCA was computed using the prcomp() function in Base R (http://www.rproject.org/) and visualised using the factoextra package. The first 2 PC were plotted, as these accounted for 93% variance. The loadings were added in their respective locations, and isolates were coloured based on resistance mechanism.

## qRT-PCR

Total RNA was extracted using the FastRNA pro blue kit (MP Biomedicals). Cells were grown to an $OD_{600nm}$ of approx. 0.4, and 2 volumes of RNA protect (Qiagen) were added. Cells were incubated for a further 5 min, before harvesting via centrifugation. Cell pellets were stored at −80˚C. Pellets were resuspended in RNA Pro solution and transferred to a tube containing lysing matrix B (MP Biomedicals). Cells were lysed via FastPrep (2 cycles of 20 s, 6 m/s) and centrifuged (16,200 × g, 4˚C, 10 min) to remove insoluble cell debris. The supernatant was transferred to a microfuge tube, and 300 μl chloroform (Sigma) was added. Samples were centrifuged again (16,200 × g, 4˚C, 15 min), and the supernatant was precipitated at 20˚C overnight after addition of 500 μl 100% ethanol. After precipitation, RNA was harvested by centrifugation (16,200 × g, 4˚C, 15 min), washed with 70% ethanol, and dried. Precipitated RNA was resuspended in 50 μl nuclease-free water, residual DNA was removed using the Turbo DNA-free kit (Invitrogen), and the RNA was cleaned and concentrated with the RNeasy Minelute cleanup kit (Qiagen), as per manufacturers' instructions.

cDNA was generated using Superscript III (Invitrogen), and 5 μg RNA was mixed with 2 μl dNTP mix and 1 μl 100 mM random primer (Eurofins), heated at 65˚C for 5 min, and cooled on ice; 8 μl 5× buffer, 2 μl 0.1 M DTT (Invitrogen), 1 μl RiboLock RNase inhibitor (Thermo Scientific), and 2 μl Superscript III were added, before incubation at 25˚C (5 min), 50˚C (30 min), and 70˚C (15 min). cDNA was adjusted to 40 ng/μl. RT negative controls were made as above, without presence of Superscript III.

Expression was measured against an exact copy number control, via standardisation with a plasmid containing 1 copy of each target DNA sequence [63] and normalised against expression of the housekeeping gene *rpoA*. Plasmids pJEB029 (containing approximately 200-bp target gene fragments of *dacS*, *dacR*, *dacJ*, and *rpoA*) and pJEB032 (containing approx. 200-bp target gene fragments of *vanR*, *vanS*, *vanG*, *vanXY*, *vanT*, and *rpoA*) were synthesised with the pUC-GW-Kan backbone (Genewiz). These were purified using the GeneJET plasmid miniprep kit (Thermo Fisher) and linearised using NdeI (NEB) and diluted to known copy number ($2 \times 10^8$ per μl). A qPCR mastermix was assembled, containing 25 μl SYBR Green JumpStart Taq ReadyMix (Sigma), 7 μl $MgCl_2$ in buffer (Sigma), forward and reverse primers (concentration determined by prior optimisation) and nuclease-free water up to 45 μl. Approximately 5 μl of cDNA (40 ng/μl), 5 μl of RT negative control, or 5 μl qPCR template plasmid pJEB029 or pJEB032 (diluted serially in lambda DNA (Promega)) were added, and qPCR was performed (BioRad CFX Connect Real Time System). Copy numbers were calculated using BioRad CFX manager (v3.1), and data were analysed in Microsoft Excel (2016) to generate copies per 1,000 copies of *rpoA*. Data were graphed and statistically analysed in GraphPad Prism (v9.0.2).

## Fluorescence microscopy

To compare vancomycin binding in different genotypes, *C. difficile* strains were grown to an $OD_{600nm}$ of 0.3 before incubating with 3 μg/ml vancomycin-BODIPY FL conjugate (Thermo Fisher) for 30 min, and 1 ml of labelled cells were then fixed by addition of 124 μl fixation solution (20 μl 1 M $NaPO_4$ (pH 7.4), 100 μl 16% (w/v) paraformaldehyde, and 4 μl 25% (w/v) glutaraldehyde) for 15 min at 37˚C. Cells were harvested by centrifugation (2 min, 6,000 × g), washed 5 times in PBS, and then mounted on an agarose pad (2% w/v) that was then inverted onto a 35-mm dish with #1.5 glass coverslip on the bottom (Ibidi). Phase contrast and fluorescent images were then collected using a 100× Plan Apochromat phase contrast objective on a Nikon Ti-E Eclipse widefield imaging microscope with Perfect Focus system using NIS Elements software, a Hamamatsu Flash 4.0 v2 camera, CoolLED PE-4000 fluorescent light source, and a Chroma filter set (49003).

Individual cells were segmented in the phase contrast channel using Ilastik [64] and resulting binary images were then eroded using Fiji [65] to better separate cells in close proximity to one another. This output was then processed through FAST [66] to obtain the centroid, orientation, and major axis length of each segmented object in the phase contrast image. Objects whose area was larger than 50 μm$^2$ and smaller than 4 μm$^2$ were excluded to remove mis-segmented tightly clustered cells and small detritus, respectively, from subsequent analyses. Segmentation data from the phase contrast images was then used to locate each cell on the corresponding fluorescent image and calculate its fluorescent intensity along its long axis. This was accomplished using an analysis methodology analogous to that previously described [67] that was implemented in Matlab (vR2023a).

To identify the fraction of cells that displayed increased vancomycin labelling at the midcell, the maximum fluorescent intensity in the middle half of the cell (between ¼ and ¾ along the cell's normalised length) was calculated. As subtle changes in focus or position within the image can lead to variations in the absolute fluorescent intensity, the maximum intensity at the centre of the cell was normalised by dividing it by the average fluorescent intensity found within the cell on either side of this central region (spanning from [⅙—¼] and [¾—⅚] along the cell's normalised length). The regions of the cell closest to the poles (i.e., [0 - ⅙] and [⅚ - 1] along the cell's normalised length) were not included in this normalisation as the fluorescent intensity tends to fall off in this region. After normalisation, the fluorescent intensity along the length of the cell was smoothed to reduce pixel noise before calculating the maximum fluorescent intensity in the middle half of the cell. The resulting normalised maximum intensity thus measures the factor by which fluorescent intensity at the centre of the cell is increased relative to that of a baseline measured elsewhere in the cell. Lastly, cells located close to clumps of unbound vancomycin, which is much brighter that the vancomycin bound to cells and thus readily identifiable, were removed from the analysis. This process yielded the normalised maximum intensity for 109 to 182 cells from each of the 5 different labelled strains and unlabelled control. The normalised maximum intensity threshold that discriminated cells with bound vancomycin at their centreline was identified by direct visual inspection and each of the images was visually checked to ensure it generated consistent results.

## Structural modelling

DacS and VanS were modelled as dimers using AlphaFold [26] and the resulting output files were visualised using ChimeraX [68].

## Statistics

Statistical analysis was performed in Graphpad Prism (v9.0.2), and $P < 0.05$ was considered significant. Data are presented as mean ± SD, unless otherwise stated. Differences in growth were analysed using the R package GrowthCurver outputs. A cross-correlation was performed in RStudio using Hmisc and corrplot packages [69,70], which determined area under curve (AUC-E) to be the most representative measure of growth. The differences in AUC-E, compared to the control strain curves, were calculated using *t* tests with Welch's correction. To test differences in sporulation efficiency, AUC was chosen as a representative measure of sporulation across all time points. AUC was calculated in GraphPad Prism (v9.0.2), and the difference between AUC for P30 isolates was compared with the WT using a one-way ANOVA with Brown–Forsythe and Welch's correction. Statistical analysis of cell length was performed in GraphPad Prism (v9.0.2). Fiji (v2.9.0) MicrobeJ (v5.13l) cell length outputs for P30 isolates were compared to the WT using a one-way ANOVA with Brown–Forsythe and Welch's correction to test for differences in cell length. To determine whether differences in gene

expression between strains were statistically significant, a two-way ANOVA with the Tukey–Kramer test was performed in GraphPad Prism (v9.0.2). In the experiments that quantified vancomycin-BODIPY binding, the function "fishertest" in Matlab (vR2023a) was used to compare the different strains/treatments using a Fisher's exact test.

## Supporting information

**S1 Fig. Growth of evolved endpoint clones.** Growth over time in rich media (TY broth) was measured at 600 nm in a 96-well microplate spectrometer. Growth of each endpoint clone (coloured lines) was compared to its matched control (black lines). Shown are the mean and standard deviation of repeats, assayed at minimum in biological and technical triplicate. For each strain, area under the curve was determined using the GrowthCurver R package and these were compared using Student's *t* tests with Welch's correction, with the *P*-value shown on each graph. All pairwise differences are significant. The data underlying this figure can be found in S5 Data.
(TIF)

**S2 Fig. Sporulation of evolved endpoint clones.** Sporulation efficiencies of each endpoint clone (coloured lines) were compared to the parental R20291Δ*PaLoc* (black lines). Stationary phase cultures were incubated anaerobically for 5 days with samples taken daily to enumerate total colony-forming units (CFUs, dotted lines) and spores (solid lines), following incubation at 65°C for 30 min to kill vegetative cells. Shown are the mean and standard deviations of biological duplicates assayed in triplicate. For each strain, spore CFU area under the curve was determined using Graphpad Prism and these were compared using Dunnett's T3 multiple comparisons test with the adjusted *P*-value shown on each graph. * = significant difference, N.S. = not significant. The data underlying this figure can be found in S5 Data.
(TIF)

**S3 Fig. Cell morphology of evolved endpoint clones.** Phase contrast light microscopy of mid-log cultures of each wild-type (**A**) and hyper-mutating (**B**) endpoint clone, with R20291Δ*PaLoc* for comparison. Shown is a representative field of view for each strain. (**C**) Imaging was performed on biological triplicate cultures and images were analysed using MicrobeJ to determine lengths of at least 185 individual cells for each strain. Shown is an all point violin plot with the median indicated by a solid horizontal line. Statistical significance of evolved isolates against the R20291Δ*PaLoc* control was calculated using a one-way ANOVA with Dunnett's T3 multiple comparisons test, ** = $P < 0.0001$, N.S. = not significant. The data underlying panel **C** can be found in S5 Data.
(TIF)

**S4 Fig. Genomic location of gene variants over time.** Accumulation of variants in the hyper-mutating *C. difficile* lineages Bc7 (**A**), Bc8 (**B**), Bc9 (**C**), Bc10 (**D**), and Bc11 (**E**). Each circle plot represents the 4.2 Mb genome of a single evolving population after 10 (inner ring), 20 (middle ring), and 30 passages (outer ring), with the locations of non-synonymous within gene variants indicated with black circles and the penetrance of each mutation in the population indicated by the size of the circle. The line graphs show the frequency of all variants (intergenic, synonymous, non-synonymous, and nonsense) in each population. The vancomycin MIC for each population is also indicated by the shaded region. Mutations also identified in the respective end point clone (Fig 1C) are highlighted by the coloured lines. The data underlying this figure, including a full list of all variants shown here, can be found in S3 Data.
(TIF)

**S5 Fig. Fixation of mutations during evolution.** Shown are the individual variants which fix (>95% penetrance) in all 10 parallel populations during vancomycin resistance evolution after 10 (**A**), 20 (**B**), and 30 passages (**C**). The frequency of each variant within their respective population is shown and the genes affected at each time point are shown in the table on the right. The genes in the *dacJRS* cluster are highlighted in yellow and *vanT* (*CD1526*) in green. The range of vancomycin MICs observed across all populations (lowest, light grey; highest, dark grey) is indicated by the shaded areas. The data underlying this figure can be found in S3 Data. (TIF)

**S6 Fig. KEGG pathway analysis of genes identified in evolved populations.** Genes impacted by mutations during evolution (except those in the transiently hyper-mutating Bc1 P20) were visualised in KEGG (Kyoto Encyclopedia of Genes and Genomes) colour mapper and assigned to cellular pathways. Two-component systems and ABC transporters were the best-represented functional classes. The data underlying this figure can be found in S5 Data. (TIF)

**S7 Fig. Growth of Bc1-recapitulated strains.** Growth over time in rich media (TY broth) was measured at 600 nm in a 96-well microplate spectrometer. Growth of endpoint clone Bc1, R20291Δ*PaLoc dacS*c.714G>T (*dacS*\*), 1,197,357_1,197,400del (*0979*\*), and vanSc.367_396dup (*vanS*\*) single, double, and triple mutants (coloured lines) were compared to R20291Δ*PaLoc* (black lines). Shown are the mean and standard deviation of repeats, assayed at minimum in biological and technical triplicate. For each strain, area under the curve was determined using the GrowthCurver R package and these were compared using Student's *t* tests with Welch's correction, with the *P*-value shown on each graph. N.S. indicates differences were not significant. The data underlying this figure can be found in S5 Data. (TIF)

**S8 Fig. Sporulation of Bc1-recapitulated strains.** Sporulation efficiencies of endpoint clone Bc1, R20291Δ*PaLoc dacS*c.714G>T and the *dacS*c.714G>T 1,197,357_1,197,400del vanSc.367_396dup triple mutant (coloured lines) were compared to R20291Δ*PaLoc* (black lines). Stationary phase cultures were incubated anaerobically for 5 days with samples taken daily to enumerate total colony forming units (CFUs, dotted lines) and spores (solid lines), following incubation at 65°C for 30 min to kill vegetative cells. Shown are the mean and standard deviations of biological triplicates assayed in triplicate. For each strain, spore CFU area under the curve was determined using Graphpad Prism and these were compared using Dunnett's T3 multiple comparisons test with the adjusted *P*-value shown on each graph. * = significant difference, N.S. = not significant. The data underlying this figure can be found in S5 Data. (TIF)

**S9 Fig. Growth and sporulation of Bc8/9-recapitulated strains.** (**A**) Growth over time in rich media (TY broth) was measured at 600 nm in a 96-well microplate spectrometer. Growth of R20291Δ*PaLoc* dacSc.548T>C and endpoint clones Bc8 and Bc9 (coloured lines) were compared to matched controls (black lines). Shown are the mean and standard deviation of repeats, assayed at minimum in biological and technical triplicate. For each strain, area under the curve was determined using the GrowthCurver R package and these were compared using Student's *t* tests with Welch's correction, with the *P*-value shown on each graph. All pairwise differences were significant. (**B**) Sporulation efficiencies of R20291Δ*PaLoc dacS*c.548T>C and endpoint clones Bc8 and Bc9 (coloured lines) compared to the parental R20291Δ*PaLoc* (black lines). Stationary phase cultures were incubated anaerobically for 5 days with samples taken daily to enumerate total colony forming units (CFUs, dotted lines) and spores (solid lines), following incubation at 65°C for 30 min to kill vegetative cells. Shown are the mean and standard

deviations of biological triplicate assayed in triplicate. For each strain, spore CFU area under the curve was determined using Graphpad Prism, and these were compared using Dunnett's T3 multiple comparisons test with the adjusted *P*-value shown on each graph. * = significant difference, N.S. = not significant. The data underlying this figure can be found in S5 Data. (TIF)

**S1 Data. Vancomycin MICs of individual endpoint isolates from evolved populations.** (XLSX)

**S2 Data. All genetic variants identified in evolved endpoint isolates.** (XLSX)

**S3 Data. All genetic variants identified in evolving populations after 10, 20, and 30 transfers.** (XLSX)

**S4 Data. Additional mutations identified by nanopore sequencing of endpoint isolates Bc1-5.** (XLSX)

**S5 Data. Numerical values underlying the quantitative data shown in manuscript figures.** (XLSX)

**S1 Table. Strains used in this study.** (DOCX)

**S2 Table. Primers used in this study.** (DOCX)

**S3 Table. Plasmids used in this study.** (DOCX)

## Acknowledgments

We thank the Wolfson Light Microscopy Facility at the University of Sheffield, for assistance with phase-contrast microscopy and the Bioinformatics Core, for training. We would also like to thank Joseph A. Kirk for construction of barcoding and Δ*PaLoc* plasmids and for the R20291Δ*PaLoc* strain.

## Author Contributions

**Conceptualization:** Jessica E. Buddle, Michael A. Brockhurst, Robert P. Fagan.

**Data curation:** Jessica E. Buddle.

**Formal analysis:** Jessica E. Buddle, Anne S. Williams, William M. Durham, Robert P. Fagan.

**Funding acquisition:** Michael A. Brockhurst, Robert P. Fagan.

**Investigation:** Jessica E. Buddle, Lucy M. Thompson, Anne S. Williams.

**Methodology:** Jessica E. Buddle, Rosanna C. T. Wright, Claire E. Turner, Roy R. Chaudhuri, Michael A. Brockhurst, Robert P. Fagan.

**Project administration:** Robert P. Fagan.

**Resources:** William M. Durham.

**Software:** Jessica E. Buddle, Rosanna C. T. Wright, William M. Durham.

**Supervision:** William M. Durham, Claire E. Turner, Roy R. Chaudhuri, Michael A. Brockhurst, Robert P. Fagan.

**Visualization:** Jessica E. Buddle, Robert P. Fagan.

**Writing – original draft:** Jessica E. Buddle, Anne S. Williams, William M. Durham, Michael A. Brockhurst, Robert P. Fagan.

**Writing – review & editing:** Jessica E. Buddle, Lucy M. Thompson, Anne S. Williams, Rosanna C. T. Wright, William M. Durham, Claire E. Turner, Roy R. Chaudhuri, Michael A. Brockhurst, Robert P. Fagan.

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
