## [Editor Report · Decision Letter 0]

20 Oct 2023

Dear Dr. Fagan, 

Thank you for submitting your manuscript entitled "Multiple evolutionary pathways lead to vancomycin resistance in Clostridioides difficile" for consideration as a Research Article by PLOS Biology.

Your manuscript has now been evaluated by the PLOS Biology editorial staff, as well as by an academic editor with relevant expertise, and I am writing to let you know that we would like to send your submission out for external peer review.

Once your full submission is complete, your paper will undergo a series of checks in preparation for peer review. After your manuscript has passed the checks it will be sent out for review. To provide the metadata for your submission, please Login to Editorial Manager (https://www.editorialmanager.com/pbiology) within two working days, i.e. by Oct 22 2023 11:59PM.

Kind regards,

Paula

---

Senior Editor

PLOS Biology

---

## [Decision Letter · Decision Letter 1]

21 Dec 2023

Dear Dr Fagan,

Thank you for your patience while your manuscript "Multiple evolutionary pathways lead to vancomycin resistance in Clostridioides difficile" was peer-reviewed at PLOS Biology. Your manuscript has been evaluated by the PLOS Biology editors, an Academic Editor with relevant expertise, and by three independent reviewers.

As you will see in the reviewer reports, which can be found at the end of this email, although the reviewers find the work potentially interesting, they have also raised a substantial number of important concerns. Based on their specific comments and following discussion with the Academic Editor, it is clear that a substantial amount of work would be required to meet the criteria for publication in PLOS Biology. However, given our and the reviewer interest in your study, we would be open to inviting a comprehensive revision of the study that thoroughly addresses all the reviewers' comments. Given the extent of revision that would be needed, we cannot make a decision about publication until we have seen the revised manuscript and your response to the reviewers' comments. Your revised manuscript would need to be seen by the reviewers again, but please note that we would not engage them unless their main concerns have been addressed. 

You'll see that reviewer #1, who is from an evolutionary background, is broadly positive, and their requests are mostly for textual clarifications. However, reviewers #2 and #3, who are both very familiar with the C. difficile field, feel that more work is needed to support the conclusions of the paper and to ensure the strength of advance that we expect at PLOS Biology. Specifically, reviewer #2 points out a 2022 paper (https://www.ncbi.nlm.nih.gov/pmc/articles/PMC9210968/) on WalR-WalK which s/he thinks reduces some of the novelty here. As a result, s/he finds the lack of follow-up of the identified genes and the effects of the mutations to be disappointing. Reviewer #3 also thinks that further experiments are needed to support the claims (e.g. overexpression and KD of DacJ, confirmation that it’s a D,D-carboxypeptidase, further functional follow-up of vanT, clarification of how the fitness costs arise, etc.). 

Having discussed the reviews with the Academic Editor, they agree that the prior work on WalRK increases the need to invest additional effort into follow-up of DacJ and VanT. Just so that nothing gets lost in translation, here's a lightly edited version of the AE's specific advice:

"Rev#2's concern about the novelty of identifying the 2 component signal-transduction system, based on Wal-RK study from David Weiss' group, is valid, as are rev#2 and rev#3's concerns about the advance presented without further functional analyses. The 1st concern should be acknowledged, but reduces the impact of the study, which makes further functional analyses the more important in order to confirm the hypothesized roles of both DacJ and VanT, as well as their role in the fitness defects in the absence of vancomycin. I feel that we should insist that the authors analyse mutant alleles without the influence of possible secondary mutations (i.e. by introducing them in the 'ancestral' background and/or knocking them out/down and overexpressing them to analyse effects on resistance and fitness)."

We appreciate that these requests represent a great deal of extra work, and we are willing to relax our standard revision time to allow you 6 months to revise your study. Please email us (plosbiology@plos.org) if you have any questions or concerns, or envision needing a (short) extension.

**IMPORTANT - SUBMITTING YOUR REVISION**

*Resubmission Checklist*

*Published Peer Review*

*PLOS Data Policy*

*Blot and Gel Data Policy*

Sincerely,

Roli Roberts

Roland Roberts, PhD

Senior Editor

PLOS Biology

rroberts@plos.org

REVIEWERS' COMMENTS:

Reviewer #1:

Review Plos Biology 

Multiple evolutionary pathways lead to vancomycin resistance in Clostridioides difficile

Clostridioides difficile is an important human pathogen, for which there are very limited treatment options, primarily the glycopeptide antibiotic vancomycin. In recent years vancomycin resistance has emerged as a serious problem in several Gram positive pathogens, but high level resistance has yet to be reported for C. difficile, although it is not known if this is due to constraints upon resistance evolution in this species. The development of vancomycin resistance is investigated by means of experimental evolution. The data in this manuscript shows that high vancomycin resistance leads to considerable fitness costs, both at the level of growth rate, as well as pathogenicity traits.

General scope:

The study is timely, increased antimicrobial resistance of Gram positives is emerging, and there is an increased interest and enough technological advancement to study the development of resistance in surveillance programs.

It is therefore important to investigate what the genetic and population genetic constraints are of the potential development of high-level vancomycin resistance of C. difficile.

Findings:

The authors find alternative resistance mechanisms that lead to high-vancomycin resistance via experimental evolution. Resistance is mediated by either vanT or dacJRS, these seem to be mutually exclusive because these are not found together in evolved end-point isolates. Generally, the high resistance is traded off with considerable fitness effects, both at the level of growth rate, as well as pathogenicity traits that impact transmission.

To my opinion, the findings in this manuscript are sound. I therefore suggest only minor revisions.

Minor: 

Line 33: trains = traits

L33: ' Mutational roadmap to inform genomic surveillance' (last sentence abstract). Although I like this sentence as a sentence, I am not sure whether the outcome of the study warrants this.

This study finds several mutations via experimental evolution that lead to vancomycin resistance, which may all be interesting and relevant to the clinical situation, but unfortunately the relevance of the mutations is not so clear (yet) for the clinical C. difficile isolates. Literature relating to the described mechanisms in the clinical situation would improve the clinical relevance, and also inform better on why genomic surveillance should focus on those particular resistance mechanisms. 

I do agree that it would be good for the surveillance community to know about these here described mechanisms, but 'mutation roadmap' based on evolution experiments in a 'not so in vivo like environment' is perhaps overstating the direction that the surveillance should take.

Line 81: mutations in the pathway are predicted: not so clear here, predicted by the authors or by literature?

Line 95: 'complete deletion of 18 kb spanning 94 the entire PaLoc that includes the genes encoding both major toxins and associated 95 regulatory proteins, creating strain R20291ΔpaLoc' . 

I do understand why the authors delete this region for practical purposes. Do the authors believe that the deletion of this part would have (additional) pleiotropic effects on the developed resistance?

Line 116: Different evolutionary scenarios: Different mutations in the MutSL lineages, faster and higher adaptation. Is this due to more combinatorial power for weak effect mutations, different population genetics leading to the selection of other mutations, or just hitch-hiking?

Line 123: Significant variation in MIC within evolved populations. Do the authors have an idea how the less resistant mutants can stay in the more highly-resistant population? And how important this finding may be for the clinical situation?

Line 156: Indels and copy-number variation missing in the analysis of the vancomycin resistant mutants? Line 183, does mention deletion though. Unclear how important indels and copy-number variants are for the development of resistance.

Line 200: Evolution experiments and assays are performed under rich medium conditions. Do the authors believe that the pleiotropic effects (that potentially limit the clinical importance of the found resistance mechanisms) also affect pathogenicity traits in other settings. In other words, could it be that the fitness costs associated with resistance development, on both the level of growth and pathogenicity, would be less (or more?) severe under other conditions? 

Having said this, resistance is also tested on rich media in the clinic (in which the authors show that there is this pleiotropic cost), perhaps this also leads to an under-detection of vancomycin resistance in C.difficile (as the authors mention, there is quite some MIC differences in an evolved population, and more resistant mutants have a higher cost)?

To be clear: I am not asking for more experiments.

Line 228: possible that affect regulation.. how?

Line 270: How many fold upregulation? 4.8-94.6

Line 272: It is unclear to me how the overexpression of DacJ leads to the reduction in vancomycin binding sites. Could the authors explain?

Line 351: Delayed emergence of vanT variants - relate back to potential importance of co-occurrence with ComR?

Lines 499 and onward; analysis also included CNV and indels?

Reviewer #2:

General Comments: 

The manuscript by Buddle et al describes the development of vancomycin resistance in C. difficile during growth under antibiotic pressure. The authors demonstrate that although high level resistance is not common among clinical isolates, significant vancomycin resistance can be achieved under selection, although with fitness costs. They explore a series of mutations that are suspected to contribute to target changes within peptidoglycan that reduce the impact of vancomycin on cell wall growth. The identification of potential mutations that confer resistance to vancomycin are of great importance, given that this is the primary antibiotic therapy for CDI. However, the locus the authors expand on as previously unidentified, were characterized in a 2022 paper, reducing the novelty of the findings. Further, the manuscript does not determine the nature of these or other mutations identified in the study to give insight into the specific mechanisms that lead to increased resistance. Additional exploration of the findings could provide helpful insight into the potential development of resistance in this pathogen. 

Major comments:

A major finding reported is the identification of a genetic locus that the authors refer to as dac genes. Although the authors state that "these genes had not been previously implicated in vancomycin resistance", there appears to be a publication describing that (PMC9210968 JBac 2022). This is the previously characterized WalRK system. 

Throughout the manuscript the authors report mutations in genes, but they do not provide follow up to determine the nature of the mutations, or independently delete the genes, which limits the utility of the work. It is not clear if transcription or function are disrupted in most cases, GOF, LOF, etc. However, the authors make conjectural statements based on presumed functions of the mutations, though multiple mutations are present throughout the genome. (For example, at Ln 231-234, they mention a substitution in CD3034, suggest its function in D-aa synthesis, and draw conclusions within the results regarding the importance of vanT to resistance.) 

The authors reference "CD" numbers as locus tags. CD is the prefix for the strain 630, but these locus tags do not match that reference genome and it is not the strain the authors use. It is most likely they mean to refer to the CDR20291_ locus tags. Please clarify and amend throughout the manuscript. Also, locus tag numbers should be provided in addition to any gene names to ensure clarity. 

MIC methods should state the number of cells used as inoculum, rather than optical density with volume, which can vary by medium and strain.

Minor comments:

Growth curves should be plotted on log scale and measurements of growth cannot be negative OD values. 

Sporulation assays- According to the literature, 65C is not sufficient to consistently kill C. difficile vegetative cells. Controls that ca

---

## [Decision Letter · Decision Letter 2]

3 Jul 2024

Dear Rob,

Thank you for your patience while we considered your revised manuscript "Multiple evolutionary pathways lead to vancomycin resistance in Clostridioides difficile" for publication as a Research Article at PLOS Biology. This revised version of your manuscript has been evaluated by the PLOS Biology editors, the Academic Editor and the original reviewers.

Based on the reviews, we are likely to accept this manuscript for publication. We would also like to give you the opportunity to address reviewer's 2 comments as you consider best. Please also make sure to address the following data and other policy-related requests.

a) We routinely suggest changes to titles to ensure maximum accessibility for a broad, non-specialist readership, and to ensure they reflect the contents of the paper. In this case, we would suggest a minor edit to the title, as follows. Please ensure you change both the manuscript file and the online submission system, as they need to match for final acceptance:

"Identification of pathways to high-level vancomycin resistance in Clostridioides difficile that incur high fitness costs in key pathogenicity traits"

Please supply the numerical values either in the a supplementary file or as a permanent DOI’d deposition for the following figures:

Figure 1ABCD, 2ABCDE, 3BC, 4B, 5AB, 6, S1, S2, S3C, S4ABCDE, S5ABC, S6, S7ABCD, S8AB, S9AB

c) Please cite the location of the data clearly in all relevant main and supplementary Figure legends, e.g. “The data underlying this Figure can be found in S1 Data” or “The data underlying this Figure can be found in https://doi.org/10.5281/zenodo.XXXXX”

d) We require the output files obtained from AlphaFold for figures 3B and 5B

e) Please ensure that your Data Statement in the submission system accurately describes where your data can be found and is in final format, as it will be published as written there.

f) Per journal policy, if you have generated any custom code during the curse of this investigation, please make it available without restrictions upon publication. Please ensure that the code is sufficiently well documented and reusable, and that your Data Statement in the Editorial Manager submission system accurately describes where your code can be found.

We expect to receive your revised manuscript within two weeks. 

*Published Peer Review History*

*Press*

Sincerely,

Melissa

Melissa Vazquez Hernandez, Ph.D.

Associate Editor

PLOS Biology

REVIEWERS' COMMENTS

Reviewer #2: 

I thank the author for clarifying the genes in question, which are annotated as WalRK, but are not the WalR-WalK from the Weiss study. However, a search for CDR20291_3437-34 (i.e. the author's dacSRJ) yields a paper (Mol Microbiol 2024; 121(6):1182-99) that has characterized this system in the 630 strain (CD630_35990 = CDR20291_34370), which demonstrated the phenotype of the HK mutant (VanS/CD630_35990) in regulation of vancomycin resistance genes and sensitivity to vancomycin. 

The author states that other groups have performed negative controls in past sporulation experiments, so they do not. Experiments without controls are not sufficiently rigorous to meet the standards for publication.

Bacterial growth should be graphed on a log scale with only a positive y-axis, even when a convenient software program cannot be used to do so with your plate reader. Again, growth cannot be negative and the raw data do not accurately portray logarithmic growth. 

I do not understand the insistence of the author to refer to the genes as "CD____" throughout the paper when these designations refer to different genes in a different strain. This is going to cause unnecessary confusion. 

Fig S7- The growth observed for the parent strain in this medium is less than half of what has been observed in other publications, which is concerning. 

Fig S8- Why are there spores in the initial cultures of the sporulation assay? The identity of the mutants shown is not clear from the nomenclature used. 

Reviewer #3:

In this revised manuscript, Buddle and colleagues investigate the genetic basis of evolved vancomycin resistance in C. difficile using experimental evolution. In their revised manuscript and response to reviewers, the authors have addressed my major concerns through the inclusion of new experimentation, analyses, and clarification within the methods and manuscript text. The authors also highlight limitations to their analysis that provide helpful context for the reader. The manuscript, which I was enthusiastic about initially, is much improved and will be of great interest to the field.

---

## [Editor Report · Decision Letter 3]

9 Jul 2024

Dear Rob,

Thank you for the submission of your revised Research Article "Identification of pathways to high-level vancomycin resistance in Clostridioides difficile that incur high fitness costs in key pathogenicity traits" for publication in PLOS Biology. On behalf of my colleagues and the Academic Editor, Arjan de Visser, I am pleased to say that we can in principle accept your manuscript for publication, provided you address any remaining formatting and reporting issues. These will be detailed in an email you should receive within 2-3 business days from our colleagues in the journal operations team; no action is required from you until then. Please note that we will not be able to formally accept your manuscript and schedule it for publication until you have completed any requested changes.

PRESS

Sincerely, 

Melissa

Melissa Vazquez Hernandez, Ph.D.

Associate Editor

PLOS Biology
